# Regulation of signaling directionality revealed by 3D snapshots of a kinase: regulator complex in action

Felipe Trajtenberg[1], Juan A Imelio[1], Matías R Machado[2], Nicole Larrieux[1], Marcelo A Marti[3], Gonzalo Obal[4†], Ariel E Mechaly[1], Alejandro Buschiazzo[1,5]*

[1]Laboratory of Molecular and Structural Microbiology, Institut Pasteur de Montevideo, Montevideo, Uruguay; [2]Biomolecular Simulations, Institut Pasteur de Montevideo, Montevideo, Uruguay; [3]Departamento de Química Biológica e IQUIBICEN-CONICET, Facultad de Ciencias Exactas y Naturales, Universidad de Buenos Aires, Buenos Aires, Argentina; [4]Protein Biophysics Unit, Institut Pasteur de Montevideo, Montevideo, Uruguay; [5]Département de Microbiologie, Institut Pasteur, Paris, France

**\*For correspondence:** alebus@pasteur.edu.uy

**Present address:** [†]Molecular Mechanisms of Membrane Transport, Institut Pasteur, Paris, France

**Competing interests:** The authors declare that no competing interests exist.

**Abstract** Two-component systems (TCS) are protein machineries that enable cells to respond to input signals. Histidine kinases (HK) are the sensory component, transferring information toward downstream response regulators (RR). HKs transfer phosphoryl groups to their specific RRs, but also dephosphorylate them, overall ensuring proper signaling. The mechanisms by which HKs discriminate between such disparate directions, are yet unknown. We now disclose crystal structures of the HK:RR complex DesK:DesR from *Bacillus subtilis*, comprising snapshots of the phosphotransfer and the dephosphorylation reactions. The HK dictates the reactional outcome through conformational rearrangements that include the reactive histidine. The phosphotransfer center is asymmetric, poised for dissociative nucleophilic substitution. The structural bases of HK phosphatase/phosphotransferase control are uncovered, and the unexpected discovery of a dissociative reactional center, sheds light on the evolution of TCS phosphotransfer reversibility. Our findings should be applicable to a broad range of signaling systems and instrumental in synthetic TCS rewiring.

## Introduction

Perception of environmental and intracellular cues is an essential feature of life. Signaling pathways enable cells to regulate genetic and biochemical programs for adaptation and survival. Two-component systems (TCSs) play a particularly important role among protein machineries that cells have evolved to carry out signaling, widely distributed in bacteria, fungi and plants. The simplest TCSs comprise a sensory histidine kinase (HK) and a response regulator (RR) component (*Gao and Stock, 2009*). The signal is transmitted downstream from HK to RR, through an orderly sequence of conformational rearrangements coupled to phosphoryl-transfer reactions. The HK is turned on or off after signal-dependent allosteric rearrangements, which control autophosphorylation on a conserved histidine residue. The phosphoryl group is then transferred from the phosphorylated kinase to a conserved aspartate on the RR receiver domain, activating it to effect a specific output response. When auto-kinase activity is turned off, HKs often act as a phosphatase of their cognate phosphorylated RR (P~RR) (*Goulian, 2010*), contributing to shutting down the pathway. Despite being paradoxical for a protein kinase, the phosphatase activity of HKs is physiologically relevant ensuring robust homeostatic responses (*Schramke et al., 2016*).

HKs are homodimeric proteins, comprising an N-terminal sensor domain and a more conserved catalytic core. Trans-membrane HKs display the catalytic region in the cytoplasm and the sensory domain within the lipid bilayer or toward the extracellular/periplasmic space. The catalytic core typically includes a helical Dimerization and Histidine phosphotransfer (DHp) domain, followed by a Catalytic and ATP-binding (CA) domain. HKs are classified according to DHp sequence signatures in HisKA, HisKA_2, HisKA_3, HWE_HK and His_kinase families. CheA-like HKs instead harbor the reactive His within a Histidine Phosphotransfer (HPt) domain that is not involved in dimerization. Overall sequence conservation among HKs can be very low, but the entire catalytic region is remarkably conserved at the structural level (*Zschiedrich et al., 2016*), suggesting that functional mechanisms are shared. RRs comprise a receiver domain (REC), which may be adjacent to additional effector domains (DNA binding, enzymatic, etc.). REC phosphorylation stabilizes structural rearrangements associated with effector domain activation (*Gao and Stock, 2010*).

Phosphorelays are examples of more complex TCS pathways, involving additional intermediate phosphotransfer proteins, such as extra REC, HPt and/or modified DHp domains. Phosphorelays allow for more complex signaling circuits and sharper regulation (*Goulian, 2010*). HKs have evolved to catalyze irreversible phosphotransfer reactions (*Porter et al., 2008*; *Potter et al., 2002*) ensuring a unidirectional input→output information flow (*Bourret and Stock, 2002*). However, evolutionary pressure has also resulted in signaling pathways where reversible steps are simultaneously required, mostly in phosphorelays (*Burbulys et al., 1991*; *Janiak-Spens et al., 2005*), where dedicated HPt modules act as phospho-donors and -acceptors. The molecular bases underlying such differential reversibility patterns remain unknown, raising the question of how signaling pathway directionality is ensured, from input signal to adaptive response.

To address this question we have used the TCS DesK-DesR from *B. subtilis* as a model (*de Mendoza, 2014*). We had previously shown that the HK DesK, a member of the HisKA_3 family, undergoes important conformational changes to switch between phosphatase- and phosphotransfer-competent states (*Albanesi et al., 2009*). The cytoplasmic portion of DesK (DesKC) including its entire catalytic region, displays a symmetric and rigid structure in the phosphatase state. DesK activation implies its rearrangement into an auto-kinase competent form with substantially higher flexibility. Upon auto-phosphorylation DesKC adopts a strongly asymmetric conformation (*Albanesi et al., 2009*) able to transfer the phosphoryl group to DesR, its downstream RR partner. The DHp α-helices include a conserved membrane-proximal coiled-coil motif, upstream of the phosphorylatable His. Such DHp segment is critical for signal transmission, folding into a coiled-coil in the phosphatase state, whereas in the phosphotransferase form this coiled-coil breaks apart (*Saita et al., 2015*). We now report the crystal structures of the DesKC:DesR complex, trapped in the phosphatase and the phosphotransferase functional states. Extensive conformational rearrangements reveal how the two reaction centers are remodeled. Our data indicate that the relative orientation and distance of the reactive histidine with respect to the receiver aspartate is a molecular determinant controlling signal directionality.

## Results

### Two crystal structures of the DesKC:DesR complex represent snapshots of the dephosphorylation and the phosphotransfer reactions

To grasp the molecular determinants of unidirectional TCS signaling, the crystal structures of DesKC in complex with DesR were determined in two functional states (*Figure 1*, *Table 1*). One using the phosphatase-constitutive mutant DesKC$_{STAB}$, and the other with DesKC$_{H188E}$ carrying a phosphomimetic substitution (*Albanesi et al., 2009*), both in complex with the REC domain of DesR. DesKC$_{STAB}$ includes amino acid replacements Ser150Ile, Ser153Leu and Arg157Ile in the DHp domain (*Figure 1—figure supplement 1A*), designed to stabilize the coiled-coil region (*Saita et al., 2015*).

### The DesKC$_{STAB}$:DesR-REC complex

The asymmetric unit (ASU) of this crystal structure shows one full complex and half of a second one, which is completed through the crystallographic two-fold symmetry operator (*Figure 1—figure supplement 1B*). Several features indicate that this complex represents a snapshot of the phosphatase reaction, hereafter denominated the 'phosphatase complex'. Symmetric organizations are a hallmark

of HKs in the 'auto-kinase off / phosphatase on' states (*Albanesi et al., 2009*; *Casino et al., 2009*; *Yamada et al., 2009*). The phosphatase complex we have now crystallized indeed displays high symmetry, both in the way the REC domains associate to the HK dimer, as well as between HK monomers. The two independently refined phosphatase complexes show one dimer of DesKC$_{STAB}$ bound to two molecules of DesR-REC (*Figure 1A* and *Figure 1—figure supplement 1C*), with the latter occupying equivalent positions on either side of the HK core. The 2:2 HK:RR stoichiometry was confirmed in solution by isothermal titration calorimetry (ITC) (*Figure 1B*, *Table 2*), revealing an entropy-driven, endothermic association reaction. Size exclusion chromatography-coupled small-angle X ray scattering (SEC-SAXS) further supported the 2:2 stoichiometry (*Figure 1C*; *Tables 3* and *4*). There is also a high internal symmetry between both HK monomers, with CA domains rigidly fixed onto the central DHp and a resulting butterfly-like shape of the whole molecule. A second feature consistent with a phosphatase-competent configuration, is that one AMP-PCP (ATP analogue) moiety is bound to each CA domain, far (>27 Å) from the His$_{188(HK)}$ phosphorylation sites on the central DHp, precluding auto-kinase activity (subscripts HK and RR highlight the protein to which the indicated residues belong). Finally, the structural rearrangements triggered at high temperatures (*Saita et al., 2015*), driving DesK to its phosphatase state, have been linked to the stabilization of an N-terminal coiled-coil (*Albanesi et al., 2009*). The three point-mutations engineered within DesK helix α1, are indeed observed to stabilize a coiled-coil structure toward the N-terminus in the phosphatase complex (*Figure 1—figure supplement 1D*). Coiled-coil stabilization was devised according to previous structures of free DesKC$_{H188V}$ (*Albanesi et al., 2009*), a mutant version that maintains normal P~DesR-specific phosphatase activities in vitro and in vivo (*Albanesi et al., 2004*). Free DesKC$_{H188V}$ structures superimpose extremely well onto the HK partner of the phosphatase complex (*Figure 1—figure supplement 1E*). SEC-SAXS data further suggest that the coiled-coil conformation in the phosphatase complex was the most abundant species in solution (*Figure 1C* and *Figure 1—figure supplement 1B*; *Table 4*), involving a large α1:α1 dimerization interface of ~2100 Å$^2$ (*Figure 1A* and *Figure 1—figure supplement 1D and E*). Regarding the RR component within the phosphatase complex, the three DesR-REC molecules in the ASU display identical 'active-like' structures (*Trajtenberg et al., 2014*). BeF$_3^-$ moieties mimicking phosphoryl groups are observed bonded to each phosphorylatable Asp$_{54(RR)}$, and coordinated to Mg$^{2+}$ cations, consistent with this crystal form being a snapshot of the phosphatase reaction at its pre-dephosphorylation step.

## The DesKC$_{H188E}$:DesR-REC complex

Wild-type phosphorylated DesKC (*wt* P~DesKC) in complex with DesR variants resisted crystallization. The crystal structure was eventually solved by using the phosphomimetic DesKC$_{H188E}$ mutant in complex with DesR-REC. Three X ray diffraction datasets were obtained from DesKC$_{H188E}$:DesR-REC crystals soaked with different concentrations of MgCl$_2$ and BeF$_3^-$ (*Table 1*). The three structures are similar, showing two DesKC$_{H188E}$:DesR-REC complexes in the ASU (*Figure 1D*). A number of observations readily distinguish this complex from the phosphatase one, supporting it has captured a ground state of the P~HK→RR phosphoryl-transfer reaction, hereafter referred to as the 'phosphotransferase complex'. In the first place, the phosphotransferase complex is highly asymmetric, a typical feature of phosphorylated and kinase-active forms of HKs (*Albanesi et al., 2009*; *Mechaly et al., 2014*; *Casino et al., 2014*). One DesKC$_{H188E}$ dimer was observed bound to one DesR-REC monomer in the crystal. This 2:1 stoichiometry was further supported by ITC (*Figure 1E* and *Table 2*) and SEC-SAXS (*Figure 1F*; *Tables 3* and *4*) in solution. The symmetry is also broken within the HK dimer, with one of the two CAs interacting with the DHp helix α1 of the other protomer (*Figure 1D*). The second CA domain is free, visible in the structure due to fortuitous crystal packing contacts. Such arrangement leaves only one RR-binding site available on the HK in the phosphotransferase state, explaining the asymmetric stoichiometry. Secondly, structural superposition of the HK partner extracted from the phosphotransferase complex, with the available structure of free phosphorylated DesK (*wt* P~DesKC [PDB 3GIG]), reveals closely similar conformations (*Figure 1—figure supplement 1F*), confirming that the structure of DesKC$_{H188E}$ in complex is a reliable model of the *wt* phosphorylated species. Finally, BeF$_3^-$ was not observed bonded to Asp$_{54(RR)}$ in any of the DesKC$_{H188E}$:DesR-REC complex structures, despite testing elevated soaking concentrations. Yet, the phosphorylation sites of the RR partners are very similar compared to the phosphatase complex, allowing for the Mg$^{2+}$ cation to bind alike (the cation was observed bound only in phosphotransferase complex structures

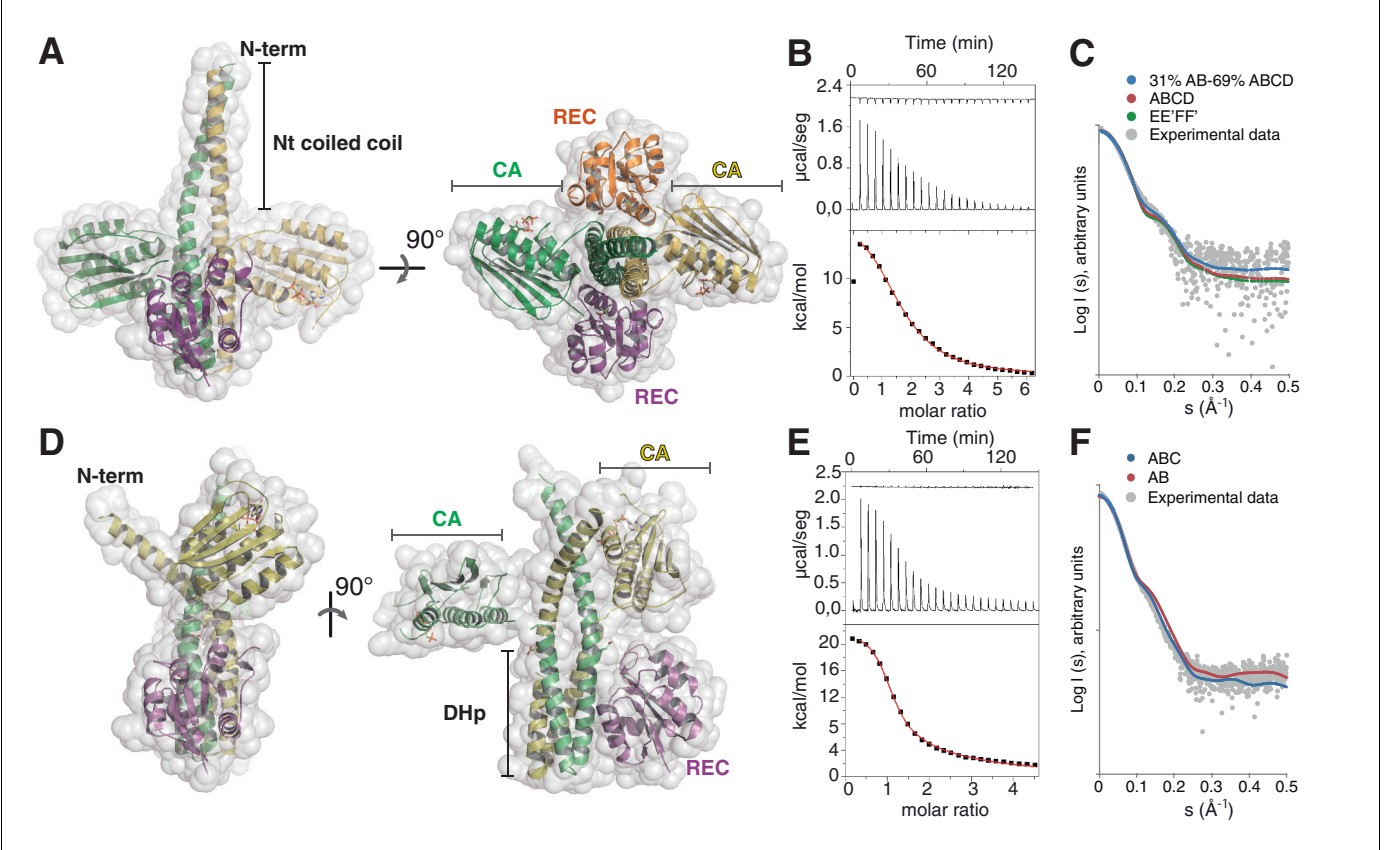

**Figure 1.** The phosphatase and the phosphotransferase complexes: crystal structures and stoichiometries in solution. (**A**) Cartoon representation of the DesKC$_{STAB}$:DesR-REC (phosphatase) complex, along two orthogonal views (left and right subpanels). The two chains within the DesKC$_{STAB}$ dimer are depicted in green and yellow, and the two DesR-REC molecules, in magenta and orange. Solvent exposed surfaces are shown in transparent gray. (**B**) Isothermal titration calorimetry (ITC) of the phosphatase DesKC$_{H188V}$:DesR-REC binding reaction, top panel showing the raw heat flow data; in the bottom, integrated heat exchange as a function of RR(monomer):HK(dimer) molar ratio. ITC was performed with the mutant DesKC$_{H188V}$, equivalently trapped in the phosphatase state (*Albanesi et al., 2009*), because DesKC$_{STAB}$ was insoluble when not co-expressed with DesR-REC. (**C**) X ray scattering curve for the DesKC$_{STAB}$:DesR-REC complex. Experimental data are plotted as gray dots, with theoretical curves overlaid as colored lines, revealing best fitting for the AB+ABCD mixture (see *Table 4*). Letters distinguish protomer chains: DesKC$_{STAB}$ dimers AB and EE' (primed labels distinguish crystallographically related partners) and DesR-REC monomers C, D, F and F'. (**D**) Cartoon representation of the DesKC$_{H188E}$:DesR-REC (phosphotransferase) complex, along two orthogonal views (left and right subpanels). Coloring scheme and solvent exposed surface displayed as in (**A**). (**E**) ITC of the DesKC$_{H188E}$:DesR-REC binding reaction, details as in (**B**). (**F**) SAXS curve for the DesKC$_{H188E}$:DesR-REC complex, details as in (**C**). Note best fitting with a one dimer DesKC$_{H188E}$ (chains AB) to one monomer DesR-REC (chain C) model (see *Table 4*).

The following figure supplement is available for figure 1:

**Figure supplement 1.** Phosphatase and phosphotransferase complexes: structural details.

soaked with high Mg$^{2+}$ concentrations). The absence of BeF$_3^-$ in the phosphotransferase state, even though Mg$^{2+}$ was eventually bound, is consistent with phosphorylated HKs not interacting with phosphorylated RR species. The molecular basis for a decreased BeF$_3^-$ affinity in the phosphotransferase became clear later by comparing both states' reaction centers (see below). Taken together, the evidence indicates the phosphotransferase complex represents a snapshot of the transfer reaction, prior to the P~His→Asp phosphoryl group migration.

## DesK and DesR interact through a 'slippery' interface

Both phosphatase and phosphotransferase complexes show that the REC domain of DesR interacts with DesK through its α1$_{(RR)}$ helix and the β5α5$_{(RR)}$ loop, with a few additional contacts from the N-terminal portion of helix α5$_{(RR)}$ and the beginning of loop β4α4$_{(RR)}$. On the HK side, the main

**Table 1.** X ray diffraction data collection and refinement statistics.

| | Phosphatase complex | Phosphotransferase complex I (low [Mg$^{2+}$]) | Phosphotransferase complex II (high [Mg$^{2+}$]) | Phosphotransferase complex III (high [Mg$^{2+}$] + BeF$_3^-$) | wt P–DesKC |
|---|---|---|---|---|---|
| **Data collection** | | | | | |
| Space group | P3$_1$21 | P2$_1$ | P2$_1$ | P2$_1$ | P3$_1$21 |
| Cell dimensions | | | | | |
| a, b, c (Å) | 94.3, 94.3, 239.9 | 87.8, 114.6, 91.6 | 87.9, 115.6, 91.6 | 88.2, 116.7, 91.9 | 94.4, 94.4, 161.8 |
| α, β, γ (°) | 90, 90, 120 | 90, 116.4, 90 | 90, 116.7, 90 | 90, 117.1, 90 | 90, 90, 120 |
| Resolution (Å) | 48.35–2.79 (2.94–2.79)* | 66.70–3.2 (3.37–3.2) | 66.78–2.9 (3.06–2.9) | 38.59–3.15 (3.32–3.15) | 36.5–3.16 (3.33–3.16) |
| Unique reflections | 31510 (4385) | 26369 (3906) | 36132 (5279) | 28033 (3944) | 14829 (2109) |
| $R_{meas}$ | 0.1 (2.07) | 0.14 (0.98) | 0.13 (1.61) | 0.1 (0.57) | 0.07 (1.18) |
| $R_{pim}$ | 0.03 (0.63) | 0.09 (0.62) | 0.05 (0.65) | 0.05 (0.29) | 0.03 (0.43) |
| CC$_{1/2}$ | 1.00 (0.56) | 0.99 (0.55) | 0.99 (0.67) | 1.00 (0.83) | 1.00 (0.77) |
| I / σI | 15.6 (1.5) | 7.7 (1.7) | 12.5 (2.4) | 11.8 (2.7) | 23.8 (1.9) |
| Completeness (%) | 99.5 (96.7) | 97.8 (98.8) | 99.1 (99.2) | 98 (95.6) | 99.9 (99.6) |
| Redundancy | 10.8 (10.3) | 2.4 (2.4) | 5.8 (6.0) | 3.8 (3.8) | 7.2 (7.2) |
| **Refinement** | | | | | |
| Resolution (Å) | 48.35–2.79 | 66.70–3.2 | 39.27–2.9 | 38.59–3.15 | 36.5–3.16 |
| Number of refls used (N in the free set) | 31454 (1568) | 26231 (1320) | 35374 (1720) | 28015 (1361) | 14787 (791) |
| $R_{work}$ / $R_{free}$ | 0.214/0.249 | 0.187/0.24 | 0.195/0.239 | 0.19/0.231 | 0.257/0.295 |
| Number of atoms | | | | | |
| Protein | 7972 | 8869 | 8702 | 8709 | 3151 |
| Ligands + ions | 93 (AMP-PCP)/6 (Mg$^{2+}$)/ 12 (MES)/12 (BeF$_3^-$)/5 (SO$_4^{2-}$) | 124 (AMP-PCP)/4 (Mg$^{2+}$)/2 (K$^+$) | 124 (AMP-PCP)/5 (Mg$^{2+}$)/2 (K$^+$) | 124 (AMP-PCP)/6 (Mg$^{2+}$)/2 (K$^+$) | 62 (AMP-PCP)/2 (Mg$^{2+}$)/6 (glycerol) |
| Water | 12 | 15 | 9 | 18 | 4 |
| B-factors (Å$^2$) | | | | | |
| Wilson plot | 98.3 | 91.4 | 95.9 | 84.1 | 128.9 |
| Mean (overall) | 118.7 [‡] | 93.1 [‡] | 102.5 [‡] | 89.6 [‡] | 123.8 [‡] |
| R.m.s. deviations | | | | | |
| Bond lengths (Å) | 0.01 | 0.01 | 0.01 | 0.01 | 0.01 |
| Bond angles (°) | 1.2 | 1.3 | 1.25 | 1.25 | 1.3 |
| Number of residues in Ramachandran plot [§] (favored / outliers) | 1025/1 | 1099/3 | 1074/2 | 1073/5 | 382/4 |
| PDB ID | 5IUN | 5IUJ | 5IUK | 5IUL | 5IUM |

*Values in parentheses correspond to the highest-resolution shell.

[‡] Including TLS contribution.

[§] Calculated with Molprobity (**Chen et al., 2010**)

element engaged in the interface is helix α1$_{(HK)}$ on the DHp domain from one DesK protomer, and minor contacts with helix α2′$_{(HK)}$ from the second protomer. This interface broadly resembles the one found in other TCS complexes (**Casino et al., 2009**; **Willett et al., 2015**; **Yamada et al., 2009**; **Zapf et al., 2000**) (**Figure 2A**).

The DHp:REC interface engages van der Waals contacts almost exclusively, including several hydrophobic residues (**Figure 2B**), which clustered among the few HK:RR covariant residue pairs (**Figure 2—figure supplement 1A**) found to be conserved in other HK and RR families

**Table 2.** Isothermal titration calorimetry parameters. For details on the two different titration procedures, see Materials and methods.

| HK:RR complexes | $K_a$ (M$^{-1}$) | $\Delta$G (x10$^3$ kcal.mol$^{-1}$) | $\Delta$H (x10$^4$ kcal.mol$^{-1}$) | T$\Delta$S (x10$^4$ kcal.mol$^{-1}$) |
|---|---|---|---|---|
| DesKC$_{H188V}$:DesR-REC | $7.7 \times 10^5$ | −7.8 | 1.5 | 2.3 |
| | $9.5 \times 10^4$ | −6.5 | 1.4 | 2.1 |
| DesKC$_{H188E}$:DesR-REC | $1.7 \times 10^6$ | −8.2 | 2.2 | 3.0 |
| | $3.0 \times 10^4$ | −5.9 | 2.1 | 2.7 |

(*Skerker et al., 2008*). Leu$_{200(HK)}$ plays a central role by inserting its side chain into a hydrophobic pocket within DesR, delimited by the side chains of Leu$_{13(RR)}$, Ala$_{16(RR)}$, Leu$_{17(RR)}$ and Leu$_{20(RR)}$ on one side, and the main chain atoms of the loop spanning residues 102–106$_{(RR)}$, on the other (*Figure 2— figure supplement 1B*). Ser$_{196(HK)}$ and Gln$_{193(HK)}$, located on the same face of α1$_{(HK)}$ as Leu$_{200(HK)}$, establish additional van der Waals contacts within a groove on the RR's surface, extending the Leu$_{200(HK)}$-lodging pocket all the way to the reaction center, which is rather open (*Figure 2—figure supplement 1B*). Very few polar contacts are observed surrounding the hydrophobic core, on both ends of the interface patch: Asp$_{203(HK)}$/Arg$_{206(HK)}$ with Ser$_{106(RR)}$/Glu$_{107(RR)}$, and a salt bridge between Arg$_{235(HK)}$ and Asp$_{103(RR)}$ on the other end. The buried surface is ~1400 Å$^2$ in the phosphatase state, the DHp:REC interface being the major contributor (~1000 Å$^2$) (*Figure 2C* and *Figure 2—figure supplement 1C*), although the CA domain adds a buried area of >400 Å$^2$. On the other hand, the phosphotransferase complex shows a slightly smaller interface (~1000 Å$^2$), dominated by the DHp: REC interaction (~900 Å$^2$) (*Figure 2D* and *Figure 2—figure supplement 1C*).

Superposing the Cα atoms of the DHp region involved in DesR-binding (DesK residues 190–235) among the nine independently refined DesKC:DesR interfaces, reveals large variability in the relative positioning of the REC domains, with rmsd as high as 2.4 Å for the RECs' 129 Cα atoms (*Figure 2E*, *Table 5*). These significant movements engage corresponding shifts in the positions of interacting residues, albeit preserving the contacts, reminiscent of a 'slippery' interface. The slippery nature of the interface is consistent with its planarity, a low number of mainly hydrophobic contacts, leading to low shape complementarity and unfilled cavities within the RR partner (*Figure 2F*).

## A single response regulator conformation is selected by the two functional states of the kinase

Comparing the phosphatase *vs* phosphotransferase complexes, the HK partner reveals substantial rearrangements. Concerning the DHp domain, a large rotational shift is observed on the membrane-proximal side of the phosphorylation site (*Figure 3—figure supplement 1A*), in contrast to the distal RR-binding site, which remains essentially invariant. The relative orientation of the CA domains is also dramatically changed (*Figure 3—figure supplement 1B*). On the other hand, the RR REC domain adopts a unique conformation in both complexes, with an average rmsd of 0.3 Å aligning all Cα atoms, and no significant contrast clustering phosphatase vs phosphotransferase structures (*Table 5*). Interestingly, this conformation adopted by the REC domain in complex with the kinase, combines structural features of both active and inactive states as seen in the free forms of RRs (*Gao and Stock, 2009*; *Trajtenberg et al., 2014*) (*Figure 3*). Namely, the β4α4 loop and β5 strand

**Table 3.** Small angle X ray scattering and derived molecular size parameters. Data derived from experiments performed by SEC-coupled SAXS. Figures for the phosphatase-trapped species DesK$_{H188V}$, are to be compared with those corresponding to HK:RR species. Full details in Materials and methods.

| Protein species | I$_{(0)}$ | I$_{(0)}$ real | Rg (Guinier) | Rg (real) | Vc | MM$_{Vc}$ (Da) | MM$_{seq}$ (Da) | Dmax (Å) |
|---|---|---|---|---|---|---|---|---|
| DesKC$_{H188V}$ | $4.47 \times 10^{-2}$ | $4.26 \times 10^{-2}$ | 31.25 | 31.21 | 442.9 | 46400 | 44780 | 100 |
| DesKC$_{STAB}$:DesR-REC | $1.16 \times 10^{-1}$ | $1.11 \times 10^{-1}$ | 31.48 | 31.42 | 494.8 | 58400 | 77110 | 103 |
| DesKC$_{H188E}$:DesR-REC | $1.05 \times 10^{-1}$ | $9.76 \times 10^{-2}$ | 32.84 | 32.18 | 486.2 | 51800 | 61790 | 103 |

**Table 4.** Fitting figures ($\chi 2$) comparing alternative theoretical SAXS curves to the experimentally collected scattering data.

| Experimental Data | Structural models | $\chi^2$ |
|---|---|---|
| | AB+ABCD mixture | 6.1 |
| | ABCD (folded coiled-coil) | 8.9 |
| | EE′FF′ (disrupted coiled-coil) | 11.6 |
| DesKC$_{STAB}$:DesR-REC | | |
| DesKC$_{H188E}$:DesR-REC | ABC | 8 |
| | AB | 24.9 |

emulate the inactive RR, while the rest of the protein resembles the free phosphorylated species. The $\beta 4\alpha 4$ loop is closed in phosphorylated RRs (e.g. PDB 4LE0), coupled to a fully wound first turn of helix $\alpha 4$ (Pro$_{85(RR)}$ displays a dihedral $\psi$ angle near $-40°$, allowing for $\alpha$-helicity). The REC molecules in both phosphatase and phosphotransferase complexes, show instead this $\beta 4\alpha 4$ loop open (with Pro$_{85(RR)}$ $\psi$ angles of respectively 126° and 130°, and the first turn of $\alpha 4$ unwound, as observed in the inactive state of the Mg$^{2+}$-free protein [PDB 4LE1]). Thus acting as a 'phosphate lid', the $\beta 4\alpha 4$ loop opens and closes to control phosphoryl-transfer chemistry on Asp$_{54(RR)}$ (*Figure 3*).

Regarding the positions of the phosphorylatable Asp, as well those of the $\beta 1\alpha 1$ loop and contiguous helix $\alpha 1$, HK-binding triggers RR 'activation', as we had previously predicted based on functional readouts and *in silico* modeling (*Trajtenberg et al., 2014*). The shift of loop $\beta 1\alpha 1$ locates key acidic residues in place to coordinate the essential Mg$^{2+}$ cation within the RR's reaction center.

Taken together, the RR adopts a single intermediate pre-activated conformation, instrumental for both dephosphorylation and phosphotransfer reactions, due to increased exposure of the phosphorylation site and stabilized configuration of the pocket poised for reaction.

## The phosphatase to phosphotransferase transition: reconfiguring the reaction center for catalysis

The crystal structure of *wt* P~DesKC was determined (*Table 1*), improving previous resolution limits (*Albanesi et al., 2009*). The reactive His$_{188(HK)}$ was observed to be phosphorylated on its N$\varepsilon$2 atom (*Figure 4—figure supplement 1A*), and superposition of this model onto the HK partner of the phosphotransferase complex (PDB 5IUK) revealed very similar structures, with 1.2 Å rmsd aligning all 1994 atoms that comprise the DHp domain and the fixed CA (*Figure 4—figure supplement 1B*). Substituting the DesKC$_{H188E}$ component in the crystal complex by the superposed *wt* P~DesKC, revealed surprisingly good geometry between P~DesKC and DesR-REC (*Figure 4—figure supplement 1C*). Energy minimization easily fixed minor interatomic bumping, rendering a model with P~His$_{188(HK)}$ well oriented toward Asp$_{54(RR)}$, poised for phosphotransfer (*Figure 4A*). The P~His$_{188(HK)}$N$\varepsilon$2, phosphorus and Asp$_{54(RR)}$O$\delta$1 atoms describe a~180° angle, with the phosphoryl group at H-bonding distance to the side chain oxygens of Thr$_{80(RR)}$ and Thr$_{81(RR)}$. The model proved to be stable in molecular dynamics simulations (*Figure 4B*), overall making chemical and biological sense. We shall thus use the minimized *wt* P~DesKC:DesR-REC model to perform further structural analyses of the phosphotransferase complex reaction center.

The phosphatase→phosphotransferase switch engages major rearrangements. An ~80° rotation of helix $\alpha 1$ is coupled to a cogwheel-like shift of helix $\alpha 2$, at the level of the phosphorylation site within the DHp domain (*Albanesi et al., 2009*). This rotation includes the phosphorylatable His$_{188(HK)}$ (*Figure 4C*), which had not been previously observed. Due to this shift, the side chain of His$_{188(HK)}$ ends up buried in the phosphatase state, establishing a H-bond with the same residue of the other protomer (*Figure 4C* and *Figure 4—figure supplement 1D*), displaced away from the reaction center as compared to the phosphotransferase (*Figure 4D*). Asp$_{189(HK)}$ right next to the phosphorylation site, follows a similarly dramatic rearrangement, well positioned only in the phosphatase complex to

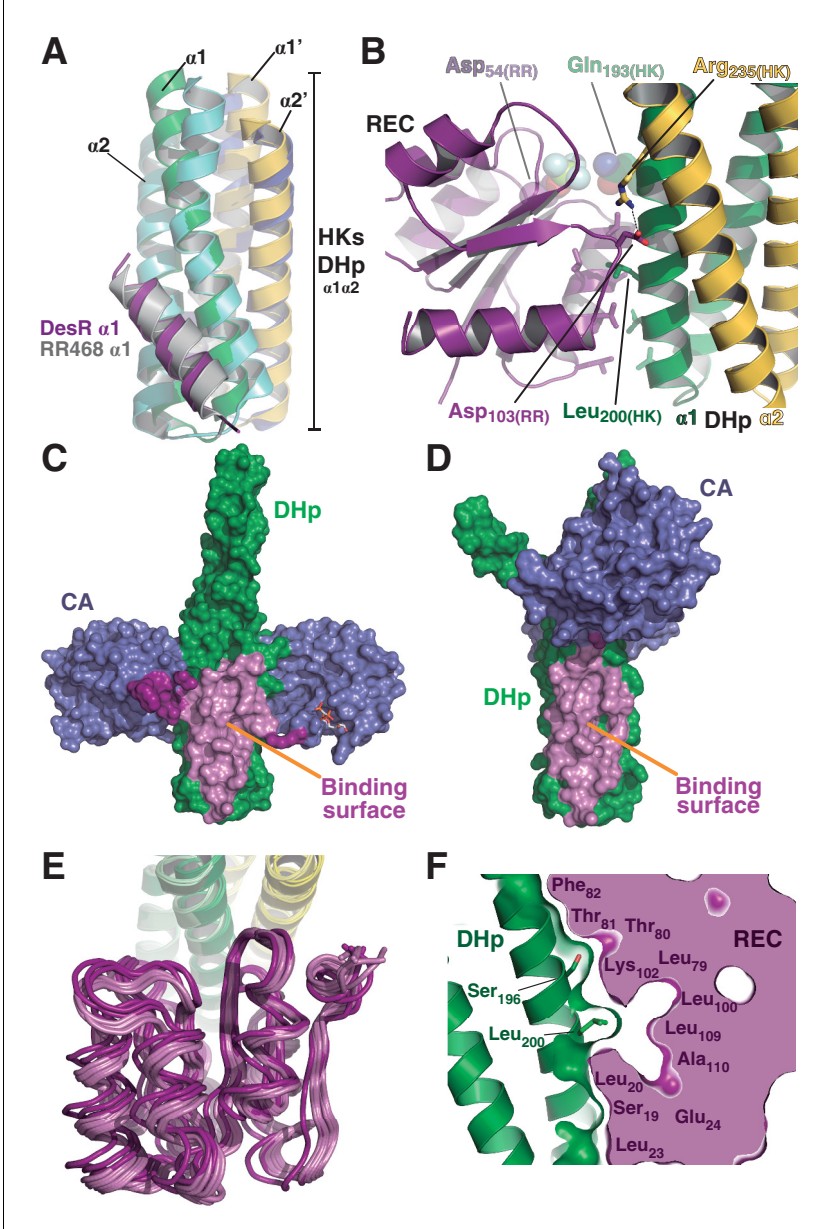

**Figure 2.** The HK:RR interface and structural variability. (**A**) Structural superposition of the DesK:DesR phosphatase complex (this study, PDB 5IUN) with HK853:RR468 from *Thermotoga maritima* (PDB 3DGE). Only selected structural elements from both partners are shown as cartoons for clarity. The DesK:DesR phosphotransferase complex is overall similar (not shown here). DesK and DesR are illustrated with the same color code as in *Figure 1*. Superposed *T. maritima* RR468 helix α1 is shown in gray, and HK853 protomers in cyan and blue. Primed labels distinguish DHp helices from each HK protomer. (**B**) Close-up of the phosphatase complex DesK:DesR interface showing only a few selected interactions for clarity (see text for detailed description). Coloring scheme as in (**A**). Note Leu$_{200(HK)}$ inserted into a hydrophobic pocket of DesR, and several other hydrophobic residues completing the interface (not labeled, shown as sticks). Among the polar contacts surrounding the hydrophobic core, Arg$_{235}$$_{(HK)}$:Asp$_{103(RR)}$ and Gln$_{193(HK)}$:BeF$_3^-$-modified Asp$_{54(RR)}$, are highlighted. (**C**) Solvent accessible surface of DesKC$_{STAB}$ indicating the interaction footprint (in magenta) with DesR in the phosphatase complex. DesK domains are highlighted in green (DHp) and blue (CAs); these domains participate differently in DesR interaction, depicted in light magenta (DHp) and dark magenta (CA). Note the ATP analogue (AMP-PCP, in sticks colored by atom) bound to the CA domains, visible in this view on the rightmost one. (**D**) Same as (**C**), but for the phosphotransferase complex, illustrating the DesR-binding surface of DesKC$_{H188E}$. (**E**) The variable position of the REC$_{(RR)}$ domain with respect to the DHp$_{(HK)}$ is illustrated, superimposing the structurally invariant region of the DHp (including the nine

*Figure 2 continued on next page*

*Figure 2 continued*

independently refined DesK:DesR complexes). The coloring scheme is the same as in (**A**), with light and dark colors distinguishing phosphotransferase and phosphatase complexes, respectively. (**F**) The solid volume corresponds to DesR-REC, which is shown sliced to highlight its outline, revealing unfilled cavities at the protein: protein interface. The relative position of interfacing amino acids is roughly indicated with residue labels. DesK DHp helices are shown in cartoon representation, with its superposed molecular surface on top. Two key residues on the HK partner are shown as sticks.

The following figure supplements are available for figure 2:

**Figure supplement 1.** Structural details of the DesK:DesR interface.

**Figure supplement 2.** Molecular dynamics (MD) simulation of the phosphotransferase complex.

---

make a salt bridge with $Arg_{84(RR)}$. Neighboring $Phe_{82(RR)}$ thus inserts its aromatic side chain into a hydrophobic pocket formed between DHp $\alpha1$ of one HK protomer and $\alpha2'$ of the other (***Figure 4D***). That the reactive histidine moves away from the catalytic site in the phosphatase state is consistent with its marginal role in P~RR dephosphorylation catalysis in HisKA_3 HKs. Indeed, the phosphorylatable His was proved nonessential for NarX-mediated P~NarL dephosphorylation (***Huynh et al., 2010***). His has been implicated in EnvZ-mediated phosphatase catalysis (***Zhu et al., 2000***) in which case its imidazole would act as a base assisting the attack of water on the P~aspartyl group (***Bhate et al., 2015***). But contradictory results cast doubts on such a direct role (***Hsing and Silhavy, 1997***; ***Skarphol et al., 1997***). Instead, a Gln in HisKA_3 (***Huynh et al., 2010***), corresponding to $Gln_{193(HK)}$ in DesK, or a Thr/Asn in HisKA HKs (***Willett and Kirby, 2012***), equivalently positioned one helical turn C-terminal to the phosphorylatable His, play a key role in the phosphatase reaction, positioning a catalytic water molecule. Such water or hydroxide molecule is well located to perform the nucleophilic attack on the phosphoryl group, as observed in the P~CheY3 phosphatase CheX (***Pazy et al., 2010***), a shared geometry among phosphatases and HKs as we now observe in the DesKC:DesR-REC phosphatase complex (***Figure 4—figure supplement 2***).

**Table 5.** Square matrix of all possible pair-wise structural superpositions. Superpositions were calculated using the nine independently refined DesKC:DesR-REC pairs, taken from the 1.5 complexes in the phosphatase asymmetric unit (named STAB1, STAB2 and STAB3) and the 2 complexes present in the ASU of each of the three different phosphotransferase structures (named as $E_{188}[1–6]$). Below the main diagonal the structurally invariant region of DesK DHp domains (residues $190–234_{(HK)}$) were superposed, whereas above the diagonal, the DesR REC domains (residues $1–129_{(RR)}$) were superposed instead. Resulting root mean squared deviations (in Å) were calculated, for both matrix halves, between all C$\alpha$ atoms of the REC domains (residues $1–129_{(RR)}$) after superposition, as indicated in each individual matrix cell. Font colors highlight the two different alignment procedures performed. The significant differences between corresponding blue- and red-colored values, indicate that the REC domains do not change within, but rather 'slip' with respect to the kinase DHp domain.

| | STAB1 | STAB2 | STAB3 | $E_{188}1$ | $E_{188}2$ | $E_{188}3$ | $E_{188}4$ | $E_{188}5$ | $E_{188}6$ |
|---|---|---|---|---|---|---|---|---|---|
| STAB1 | | 0.229 | 0.158 | 0.414 | 0.349 | 0.442 | 0.361 | 0.344 | 0.343 |
| STAB2 | 0.826 | | 0.188 | 0.496 | 0.459 | 0.512 | 0.334 | 0.343 | 0.333 |
| STAB3 | 1.362 | 1.917 | | 0.393 | 0.343 | 0.414 | 0.292 | 0.287 | 0.277 |
| $E_{188}1$ | 1.168 | 1.133 | 1.682 | | 0.275 | 0.146 | 0.305 | 0.301 | 0.301 |
| $E_{188}2$ | 0.909 | 1.140 | 1.224 | 0.584 | | 0.301 | 0.351 | 0.309 | 0.341 |
| $E_{188}3$ | 1.085 | 1.110 | 1.530 | 0.233 | 0.459 | | 0.346 | 0.315 | 0.313 |
| $E_{188}4$ | 1.548 | 1.140 | 2.453 | 0.893 | 1.316 | 1.040 | | 0.175 | 0.117 |
| $E_{188}5$ | 1.295 | 0.978 | 2.096 | 0.593 | 0.969 | 0.716 | 0.449 | | 0.140 |
| $E_{188}6$ | 1.424 | 1.058 | 2.300 | 0.770 | 1.182 | 0.905 | 0.193 | 0.306 | |

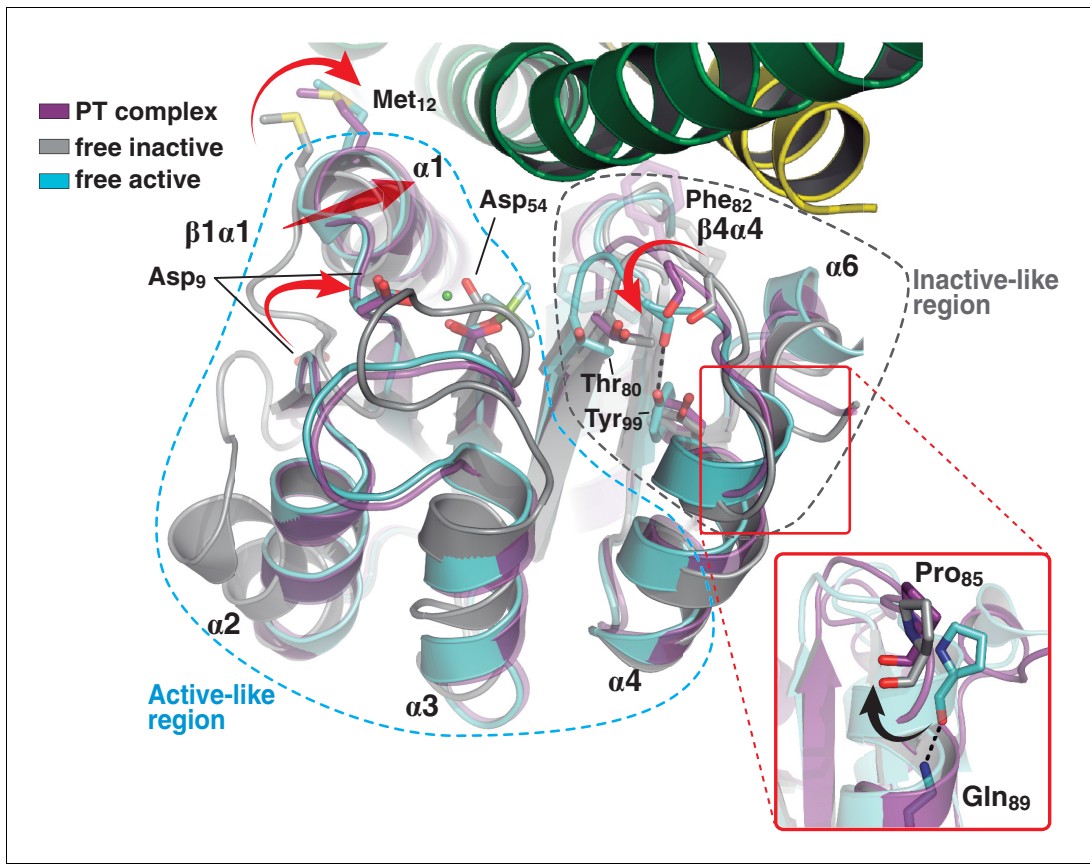

**Figure 3.** A shared RR REC domain conformation in the phosphatase and phosphotransferase complexes, has active- and inactive-like features. The phosphotransferase complex (this study, PDB 5IUK) is shown in cartoon representation, following the same coloring scheme as in *Figure 2*. REC domains from reported free DesR species are shown superposed, in gray the inactive $Mg^{2+}$-free RR (PDB 4LE1) and the active species in cyan (PDB 4LE0). Note that the complexed form of the regulator (in magenta) is mainly similar to the active species, including in the $\beta1\alpha1$ loop and $\alpha1$ helix (that play key roles in the structuring of the RR's active site), even though the RR is not phosphorylated on the reactive $Asp_{54(RR)}$. Dashed outlines highlight the regions that most resemble each functional state. Note that the inactive-like region essentially involves the end of $\beta4$ and the $\beta4\alpha4$ loop, which closes in as a phosphate lid in P~DesR. $Phe_{82(RR)}$ (shown in transparent sticks for clarity) interacts with the HK, playing a role in phosphate lid opening in both phosphatase and phosphotransferase complexes. $Tyr_{99(RR)}$ on strand $\beta5$ (the strand is not visible below $\alpha4$), typically forming a H-bond with the main chain O of $Phe_{82(RR)}$ in activated RRs, is observed farther away in both DesK:DesR complexes, yet another feature of inactive states. Inset: a close-up of the N-terminal tip of helix $\alpha4$ is shown. The dihedral $\psi$ angle of $Pro_{85(RR)}$ is linked to the active/inactive shift of the $\beta4\alpha4$ loop.

The following figure supplement is available for figure 3:

**Figure supplement 1.** Major structural differences of the HK partner, comparing the phosphatase and phosphotransferase complexes.

The rotational phosphatase→phosphotransferase rearrangement disrupts the $Asp_{189(HK)}$:$Arg_{84(RR)}$ ionic bond, and coupled to the $\alpha1_{(HK)}$ and $\alpha2'_{(HK)}$ movements, the position of $Phe_{82(RR)}$ shifts, propagating along the rest of the phosphate lid $\beta4\alpha4$ loop (*Figure 4D*). This shift accounts for the capacity of $Thr_{80(RR)}$ to be at H-bonding distance from the P~$His_{188(HK)}$ phosphoryl moiety, accompanying its transfer, and maintaining the H-bond to the phosphate on P~$Asp_{54(RR)}$ (*Figure 4D*). The side chain of $Phe_{82(RR)}$ is also contacting P~$His_{188(HK)}$, which could increase the pKa of its imidazole N$\delta$1 atom due to cation-$\pi$ interactions (*Loewenthal et al., 1992*). Indeed, P~DesKC appeared to catalyze slower phosphoryl-transfer to DesR-$REC_{F82A}$ compared to *wt* DesR-REC (*Figure 4E*), consistent with a

poorer Nδ1 protonation, and hence a stronger P-N phosphoramidate bond (*Attwood et al., 2007*). Systematic mutagenesis at RR position 82 and functional assays (catalytic activities and HK:RR affinities) will be needed to reach a definitive conclusion.

Thus, the phosphatase and phosphotransferase catalytic activities depend upon modifications in the position of a subset of residues in both partners. Both complexes share a common RR intermediate conformation, and the reaction outcome is dictated by the HK switch of helices α1 and α2.

## Dissociative phosphotransfer is linked to unidirectional signaling

DesK:DesR-catalyzed phosphotransfer occurred essentially in the forward, P~DesK→DesR direction (*Figure 5A*), extending earlier suggestions (*Albanesi et al., 2004*) into quantitative figures, and indicating that back-transfer is minimal. How is immediate P~RR dephosphorylation prevented once the phosphate has been transferred? The side chain of Thr$_{80(RR)}$ at the end of β4 strand, appears to drive the whole phosphate lid to a closed configuration, escorting the phosphoryl group during P~His→Asp migration (*Figures 3* and *4D*). To test whether a closed phosphate lid indeed plays a role in minimizing P~RR dephosphorylation, as a means to ensuring phosphotransfer unidirectionality, Phe$_{82(RR)}$ was substituted by alanine. Phe$_{82(RR)}$ covers the P~Asp$_{54(RR)}$ in active free DesR, while exposing it to bulk solvent through phosphate lid-opening when bound to DesK (*Figure 4—figure supplement 2*). The half-life of P~DesR$_{F82A}$ was significantly reduced when compared to *wt* P~DesR (*Figure 5B*), confirming that the phosphate lid is a key structural element to avoid otherwise futile P~RR dephosphorylation after phosphotransfer.

Then how is P~RR→HK back-transfer inhibited? The phosphate lid could also play a role. Yet, comparing *wt* P~DesR with point-mutants P~DesR$_{F82A}$ or P~DesR$_{R84A}$, no change was detected in their marginal abilities to transfer the phosphoryl group back to DesK (data not shown). There is however a parameter that was found to discriminate irreversible phosphotransfer systems from others that reveal appreciable reversibility: the reactional His-Asp distance. The distance between P~His$_{188(HK)}$Nε2 and Asp$_{54(RR)}$Oδ1 in the phosphotransferase complex is ~7.6 Å (*Figures 4A* and *5C*), strongly suggesting that the phosphotransfer reaction occurs through a loose (or predominantly dissociative) nucleophilic substitution mechanism. This is in contrast to what has been observed in reversible complexes (*Figure 5C*), which efficiently catalyze phosphotransfer also in the opposite, P~Asp→His direction. The latter display significantly shorter Nε2-to-Oδ1 distances, consistent with an associative nucleophilic mechanism (*Table 6*). Examples include several phosphorelay protein pairs (*Bauer et al., 2013*; *Zapf et al., 2000*; *Zhao et al., 2008*) as well as *Escherichia coli* CheY variants in complex with imidazole (*Page et al., 2015*), or yet two autophosphorylating HKs where the phosphoryl group is poised to migrate from ATP to the reactive histidine (*Casino et al., 2014*; *Mechaly et al., 2014*). In contrast, HK:RR TCSs with unidirectional P~His→Asp transfer, such as *B. subtilis* DesK:DesR (this study), *Thermotoga maritima* HK853:RR469 (*Casino et al., 2009*) and *Rhodobacter sphaeroides* CheA3:CheY6 (*Bell et al., 2010*), consistently show >6.6 Å HisNε2-AspOδ1 distances and correlated large reaction coordinate distances when these could be measured (*Table 6*). It will be interesting to test whether the phosphorelay pair ChpT:CtrA from *Brucella abortus* (*Willett et al., 2015*) displays reversible transfer, as anticipated on the basis of the reported 5.9 Å HisNε2-AspOδ1 distance.

In a loose reaction center like that of DesK:DesR, a largely dissociated meta-phosphate intermediate would be stabilized migrating toward the positive charges of the Mg$^{2+}$ cation and the conserved Lys$_{102(RR)}$ in the RR partner. The position of these positive charges, next to the RR phosphorylation site, introduces an intrinsic asymmetry within the reaction center, which must translate into unequal likelihood for phosphoryl-transfer directions. As the Mg$^{2+}$ position becomes more asymmetric, the P~Asp→His back-transfer would be less favorable, explaining unidirectionality. The ratio between the Mg$^{2+}$-Asp$_{(Oδ1)}$ and the Mg$^{2+}$-His$_{(Nε2)}$ distances, can be used to quantitate such asymmetry within reaction centers. This ratio was calculated for different TCS complexes (*Table 6*), indeed clustering those complexes that catalyze reversible P~Asp→His/P~His→Asp reactions, apart from the ones that carry out P~His→Asp unidirectional transfer (*Figure 5D*). Signaling pathways appear to have evolved associative or dissociative mechanisms, corresponding respectively to more reversible or irreversible reactions, such that they are fit to drive a defined directionality in the flow of information.

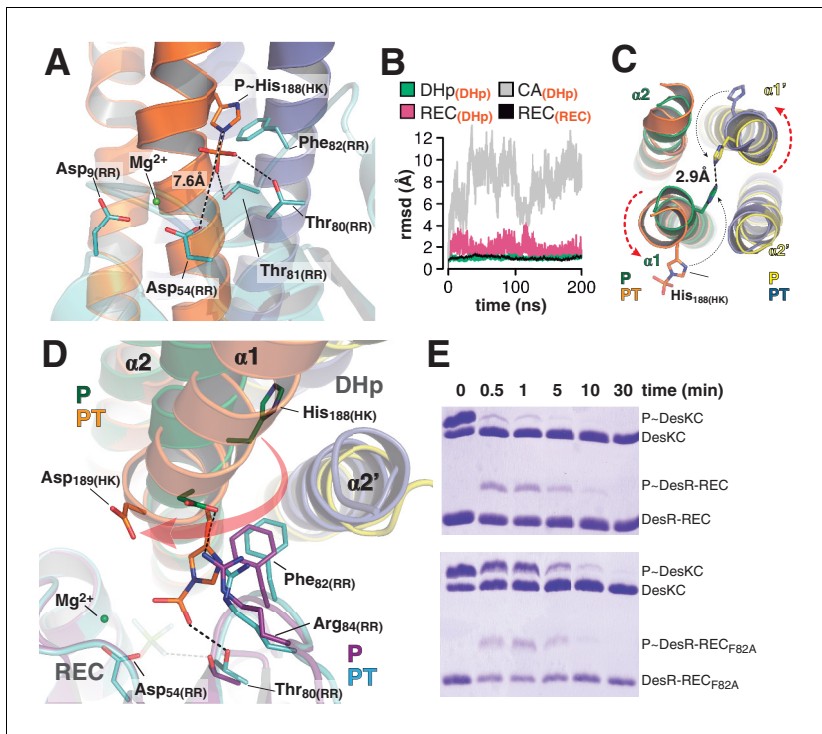

**Figure 4.** At the reaction center: the phosphatase to phosphotransferase transition. (**A**) Cartoon model of the phosphotransferase reaction center prior to DesR phosphorylation (see model construction details the Materials and methods section). DesK protomers (orange and blue) and DesR (cyan, transparent for clarity) are displayed with selected key residues in sticks colored by atom. $Mg^{2+}$ (green sphere) is already in place, coordinated by the two shown Asp residues and water molecules (not included). Phosphoryl moiety interactions with DesR, and the reactive Asp-His distance are indicated. (**B**) Evolution of the atomic coordinates of the phosphotransferase complex along molecular dynamics (MD) calculations. Selected HK or RR domains were structurally aligned (marked with orange subscripts on each curve's label on top), to thereafter compare the evolving MD model with the initial experimental structure (calculating rmsds of chosen domains as marked in black fonts on each curve's label on top). Resulting rmsds for all $C\alpha$ atoms of chosen domains are plotted (colored curves) as a function of time. Note that the time lapse is enough to detect large CA mobility (gray curve) whereas the DHp:REC complex remains attached and stable (pink curve). (**C**) Cartoon illustration of the HK DHp domains of the phosphatase complex (**P**), with its two HK protomers in green and yellow, superposed onto the phosphotransferase (**PT**) in orange and blue. Residue $His_{188(HK)}$ (in sticks) reveals the rotational rearrangement between both states. (**D**) Similar phosphatase *vs* phosphotransferase illustration as in (**C**), along a different view. The RR partner is now shown (magenta for the phosphatase complex [**P**], cyan for the phosphotransferase [**PT**]). The $Arg_{84(RR)}$:$Asp_{189(HK)}$ salt bridge is disrupted in the PT complex due to the DHp rotational shift (red arrow). Note the shift of the RR $\beta 4\alpha 4$ loop, including $Phe_{82(RR)}$, propagating toward $Thr_{80(RR)}$. The latter is positioned at H-bonding distance to the phosphoryl group either on the P~His or on the P~Asp residue (the $BeF_3^-$ moiety in the P complex is transparent), with a shift of 1.5 Å of its side chain O atom. (**E**) Phosphotransfer kinetics comparing *wt* DesR-REC (top panel) with phosphate lid mutant $DesR_{F82A}$-REC (bottom panel), revealed by Phos-tag SDS-PAGE.

The following figure supplements are available for figure 4:

**Figure supplement 1.** Electron density maps and structural similarity of wt P~DesKC compared to $DesKC_{H188E}$ in the phosphotransferase complex.

**Figure supplement 2.** Structural details of the reaction centers.

**Figure supplement 3.** Phosphate lid opening for RR dephosphorylation.

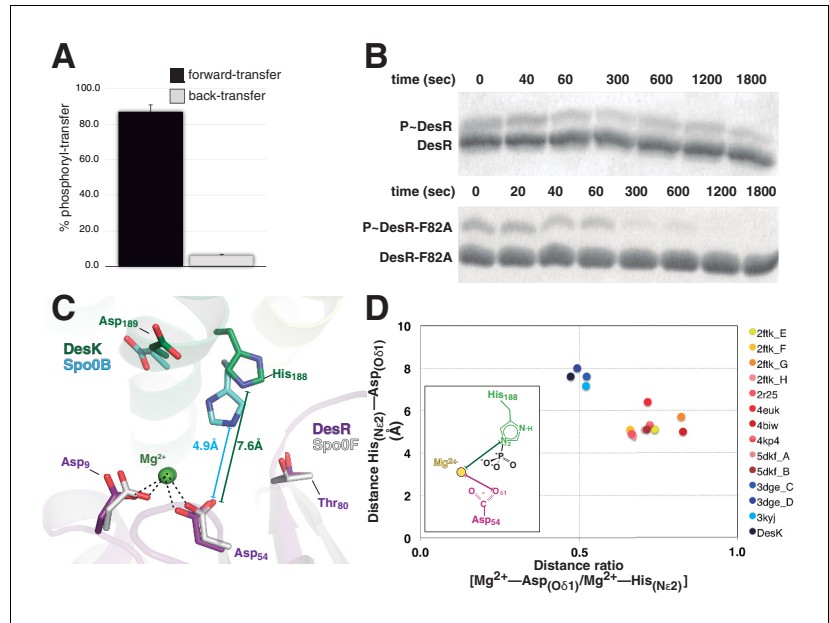

**Figure 5.** Dissociative phosphotransfer and asymmetric position of the divalent cation. (**A**) Phosphoryl-transfer reactions were analyzed in vitro using Phos-tag SDS-PAGE. Band intensities corresponding to unphosphorylated and phosphorylated species were quantified by densitometry. Forward- and back-transfer directions were compared, with bars representing the amount of phosphotransfer in each direction, plotted as the percentage of phosphorylated DesKC with respect to its total initial amount (100%). Forward phosphotransfer (black bar) was measured by incubating pre-phosphorylated P~DesKC and DesR-REC. Back-transfer (gray bar) was assayed by incubating pre-phosphorylated P~DesR and DesKC. The standard deviation for triplicate replicas are shown as error bars. (**B**) Stabilization of the phosphorylated DesR species by the phosphorylation lid. The intrinsic dephosphorylation velocities of P~DesR-REC and P~DesR$_{F82A}$-REC were compared using Phos-tag SDS-PAGE. One of three similar replicas is shown. (**C**) Structural superposition of the DesK:DesR phosphotransferase complex (colored in green:magenta) onto the *B. subtilis* Spo0B:Spo0F complex (PDB 2FTK, depicted in cyan:white), aligning the DesR and Spo0F REC domains. Selected key residues are labeled and numbered according to DesR's sequence. Note the strictly conserved position of the divalent cation (green sphere). The relative shift in the position of the phosphorylatable histidine (His$_{188}$ in DesK corresponds to His$_{30}$ in Spo0B), is highlighted by depicting the His$_{(N\epsilon2)}$-Asp$_{(O\delta1)}$ distances, not including the phosphoryl group for clarity. Note that besides the His shift, the overall DHp:REC binding interface is otherwise preserved. (**D**) Plot of His$_{(N\epsilon2)}$-Asp$_{(O\delta1)}$ distance vs degree of asymmetry in the position of the Mg$^{2+}$ cation, comparing independently refined TCS complexes (color-labeled). The inset shows the two interatomic distances used to calculate the ratio plotted in the x-axis. Note the clustering in two groups, which discriminate complexes catalyzing reversible (yellow-to-brown range of points) vs irreversible (blue range of points) phosphotransfer reactions.

## Discussion

Signaling requires *specificity*, to ensure that a given stimulus is linked to a defined adaptive response while minimizing cross-talk; *efficiency*, to avoid wasting cellular energy through futile cycles; and *directionality*, to guarantee the information is directed from the stimulus to the output response. Two component systems constitute a particularly interesting biological model to understand signaling, considering that so many histidine kinases catalyze both kinase and phosphatase reactions according to their functional status, implying exquisite regulation mechanisms at play.

By solving the crystal structures of a *bona fide* HK in complex with its cognate RR in two distinct functional states, such molecular mechanisms of TCS unidirectional signal transmission are being uncovered. These structures correspond to snapshots of the phosphotransferase reaction just prior of actual phosphoryl migration, and of the phosphatase reaction in a pre-dephosphorylation complex.

**Table 6.** Summary of reactive His$_{(HK)}$-to-Asp$_{(RR)}$ distances in reported TCS protein complexes.

| | Group (based on distance discriminants[*]) | Complex partners (N° complexes per ASU[†]) | Distance his(Nε2)-Asp(Oδ1) (Å)[‡] | Reaction coordinate distance (Å)[§] | Distance ratio [Mg$^{2+}$-Asp$_{(Oδ1)}$/ Mg$^{2+}$-His$_{(Nε2)}$] | Pdb (ID) | Phosphotransfer reaction direction | Remarks |
|---|---|---|---|---|---|---|---|---|
| *Phosphorelays* | I | Spo0F: Spo0B[¶] (4) | 5.7; 5.3; 5.1; 5.1 | 4.1; 3.8; 3.6; 3.5 | 0.82; 0.74; 0.65; 0.67 | 2FTK | P~Asp$_{(Spo0F)}$→ His$_{(Spo0B)}$ | Spo0B bears a DHp domain |
| | | SLN1: YPD1[**] (1) | 4.9 | 3.2 | 0.67 | 2R25 | P~Asp$_{(SLN1)}$→ His$_{(YPD1)}$ | YPD1 bears an HPt domain |
| | | AHK5: AHP1[††] (1) | 6.4 | ND[8] | 0.72 | 4EUK | P~Asp$_{(AHK5-REC)}$→ His$_{(AHP1)}$ | AHP1 bears an HPt domain |
| | | ChpT: CtrA[§§] (2) | 5.9[¶¶] | ND[‡‡] | 0.64[***] | 4QPJ | P~His$_{(ChptA)}$→ Asp$_{(CtrA)}$ | ChpT bears a DHp domain P~Asp$_{(CtrA)}$→His$_{(ChptA)}$ remains to be confirmed |
| *HKs undergoing auto-phosphorylation* | I | CpxA: ATP[†††] (1) | 5.0[‡‡‡] | 3.4 | 0.82 | 4BIW | ATP→His$_{(CpxA)}$ | AMP-PCP was used as non-hydrolizable ATP analogs (between phosphates β and γ there is a C atom substituting the O) |
| | | HK853/ EnvZ chimera: ATP[14] (1) | 5.3[‡‡‡] | 3.6 | 0.72 | 4KP4 | ATP→His$_{(HK853)}$ | |
| *RR-mediated phosphorylation of imidazole* | I | Imidazole: CheY[¶¶¶] (2) | 5.1; 5.1 | 3.3; 3.3 | 0.71; 0.73 | 5DKF | P~Asp$_{CheY}$ → imidazole | In vitro engineering of a phosphorelaying system from CheY to PhoR, using imidazole as a rudimentary HPt |
| *HK:RR complexes* | II | HK853: RR468[****] (2) | 7.6; 8.0[††††] | 5.7; 6.1 [††††] | 0.51; 0.48[††††] | 3DGE | P~His$_{(HK853)}$→ Asp$_{(RR468)}$ | HK:RR complex, snapshot of the phosphatase state according to the authors. |
| | | DesK: DesR[‡‡‡‡] (2) | 7.6[§§§§] | 5.8[§§§§] | 0.47[§§§§] | 5IUK | P~His$_{(DesK)}$→ Asp$_{(DesR)}$ | HK:RR complex in the phosphotransferase state |
| | | CheA3: CheY6[¶¶¶¶] (1) | 7.3 | ND[‡‡] | 0.52[*****] | 3KYJ | P~His$_{(CheA3)}$→ Asp$_{(CheY6)}$ | 3KYJ is not phosphorylated on either partner; 3KYI with P-His on CheA3 displays an unproductive P~His rotamer, otherwise confirming 3KYJ's 3D organization |

[*] Distance between reactive His(Nε2)-Asp(Oδ1) less than or greater than 6.5Å, and distance ratio [Mg2+-Asp(Oδ1) / Mg2+-His(Nε2)] less than or greater than 0.6

[†] ASU = asymmetric unit

[‡] Each distance corresponds to the one measured in each one of the independently refined complexes in the ASU.

[§] Distances are reported in the same order as in the previous column, with correspondence among same individual complexes.

[¶] Bacillus subtilis (*Varughese et al.,2006*)

[**] Saccharomyces cerevisiae (*Zhao et al., 2008*)

[††]Arabidopsis thaliana (*Bauer et al., 2013*)

[‡‡] Not determined : no phosphoryl group or phosphoryl-mimetic present in the structure.

[§§] Brucella abortus (*Willett et al., 2015*)

[¶¶] Only one of the two complexes in the ASU (ChpTchainA:CtrAchainC, display the reactive His and Asp properly oriented poised for reacting. Only one distance is thus recorded.

[***] No Mg2+ cation was actually bound on CtrA. A metal atom was modeled by superimposing the structure of RR468 with bound BeF3 (PDB 3GL9), one of the top ranking structures in multiple structural alignments with CtrA (DALI Z score 12.9, rmsd 0.8Å superimposing 98 αCs).

[†††] Escherichia coli (*Mechaly et al., 2014*)

[‡‡‡] In the autophosphorylation complexes the distance is recorded between His(Nε2) and the position of the O between phosphates β and γ.

[§§§] Thermotoga maritima (HK853) / E. coli (EnvZ) (Casino et al., 2014)

[¶¶¶] E. coli (*Page et al., 2015*)

[****] Thermotoga maritima (*Casino et al., 2009*)

†††† These distances have been calculated by superimposing the phosphorylation-mimetic structure of RR468 alone with bound BeF3 (PDB 3GL9), onto 3DGE, in order to use better estimations of the receiver Asp position as well as of the phosphorus atom.

‡‡‡‡ B. subtilis, this report.

§§§§ Glu188 present in the crystal structure was substituted by wt P~His following superposition of wt phosphorylated DesKC (PDB 5IUM) onto one of the phosphotransferase DesKCH188E:DesR-REC complexes in the ASU (chains A-B:E). Distances are reported after energy minimization. See Materials and methods.

¶¶¶¶ Rhodobacter sphaeroides (*Bell et al., 2010*)

***** The position of Mg2+ cation was modeled using PDB 4TMY as template.

## Specificity determinants within a loose and slippery interface

The small number of interactions and low surface complementarity that we observe in the several DesKC:DesR complexes (*Figure 2F*) are consistent with the observed 'slippery' nature of the protein:protein interface, displaying significant shifts in the relative positions of both partners. Further supporting that this ensemble of conformations is functionally relevant, molecular dynamics simulations of the phosphotransferase complex (*Figure 2—figure supplement 2*) recapitulated the gliding shifts of one partner with respect to the other, reaching the different arrangements observed in the crystal structures at different moments of the simulations (*Figure 2—figure supplement 2B*). Loose HK:RR association architectures can actually be observed in other TCS-related complexes (*Zapf et al., 2000*; *Casino et al., 2009*; *Willett et al., 2015*), suggesting it is a universal feature. This explains the considerable promiscuity reported in different TCSs in in vitro assays, if long enough incubation times are allowed for a given HK to phosphotransfer to different RRs, including surrogate partners (*Skerker et al., 2005*). HK:RR interactions thus ensure high specificity, but also avoiding exceedingly high stability, which would hinder proper functional modulation and, ultimately, signal transmission altogether. Evolutionarily selected to be loose, slippery interfaces are inherently tolerant for substantial sequence variation of protein:protein contacting residues. Such tolerance is consistent with the extensive degeneracy found in TCS pairs (*Podgornaia and Laub, 2015*), likely a general feature of HK:RR interactions. However, a permissive tendency for substitutions would appear to be contradictory with the high protein:protein specificity requirements of these systems. The DesK:DesR structures now provide with molecular details explaining why epistasis is found as a key property of HK:RR interfaces (*Podgornaia and Laub, 2015*). Epistatic amino acid changes are those that depend on the presence of other substitutions, leading to different effects in combination than individually. While interfacing residue replacements on any one of the TCS components can easily be accommodated due to the loose character of the protein:protein association, epistatic substitutions in the same component and/or in the other partner, are called to play essential stabilizing or destabilizing compensatory roles. Epistatic substitutions are thus selected, such that partner recognition and non-cognate discrimination are maintained, but without leading to overstable, non-functional complexes.

## A novel conformation of the response regulator when complexed with the histidine kinase

Our data indicate that the HK is largely unable to discriminate binding to RR or P~RR on the basis of structural determinants. The structures show the RR REC domain adopting identical conformations in both phosphatase and phosphotransferase complexes, presenting active- and inactive–like features in different regions of the protein, arranging the RR's active site for phosphoryl group migration (*Figure 3*). The two complexes also show equivalent HK:RR interfacing surfaces, with all major contacts preserved. Among the few differential contacts, they mainly implicate the HK's CA domain, but unlikely to grant discrimination ability given the highly similar conformation of the RR in those regions.

Calorimetry data are consistent with the crystallographic evidence, revealing only modest differences between DesR-REC:DesKC$_{H188V}$ and DesR-REC:DesKC$_{H188E}$ binding affinities (*Figure 1* and *Table 2*), further supporting that both HK functional states interact with DesR-REC with similar strengths. Collecting calorimetric data by titrating DesKC with P~DesR-REC would be interesting to directly compare with the unphosphorylated form, but binding confounds with simultaneous P~RR dephosphorylation, and the addition of BeF$_3^-$/Mg$^{2+}$ to mimic RR phosphorylation resulted in protein

aggregation. We did succeed in engineering a different phosphatase-trapped DesKC mutant, by insertion of additional heptad repeats in the coiled-coil segment (to be published elsewhere), indeed displaying equivalent DesR-REC-binding $K_D$ figures compared to DesKC$_{H188E}$. HK:RR binding data in other TCSs are scarce, particularly exploring differential affinities according to the proteins' phosphorylation status. The few cases dealing with HKs that switch between phosphatase/phosphotransferase-competent states, do provide further evidence in support of non-discrimination. EnvZ binds OmpR and P~OmpR with equivalent affinities (*Yoshida et al., 2002*), as well as PhoQ binding to PhoP or P~PhoP (*Castelli et al., 2003*). Future studies shall uncover the relevance of additional mechanisms at the cell level that might favor particular HK:RR pairs to form, such as absolute and relative TCS protein concentrations, stability of non-phosphorylated HK:RR species in complex, sub-cellular spatial dynamics of each partner, and precise time-courses for HK and RR phosphorylated species to appear/disappear.

## A HK:RR concerted switch controls efficiency along the signaling pathway

Considering all the evidence, we can now propose a conceptual model explaining the molecular workings of the DesK/DesR signaling pathway (*Figure 6*). When the pathway is turned off, HK auto-phosphorylation is inhibited by: 1) hindering CA domains mobility, holding ATP substrates far from the His$_{188(HK)}$ phosphorylation site; and, 2) DHp helical rotation, making the His$_{188(HK)}$ side chains inaccessible (*Figures 4D* and *6A*, *Video 1*). This 'open' autokinase-off configuration is linked to a folded N-terminal coiled-coil, in which DesK can bind P~DesR and trigger phosphatase catalysis. Linked to P~DesR phosphate lid opening, Asp$_{189(HK)}$ binds to Arg$_{84(RR)}$, and Phe$_{82(RR)}$ becomes inserted into a DHp pocket (*Figure 6A*, *Video 1*). Phe$_{82(RR)}$ is thus removed from the closed position, which would otherwise clash with an incoming nucleophilic water molecule (*Figure 4—figure supplements 2* and *3*) that brings about dephosphorylation (Pazy et al., 2010). DesK Gln$_{193(HK)}$ is well located to position the catalytic water to perform dephosphorylation, consistent with the essential role of the homologous Gln residues in the phosphatase reactions mediated by the phosphatase CheZ (*Zhao et al., 2002*) or by the HK NarX (*Huynh et al., 2010*). The rotational movement of His$_{188(HK)}$, placing it at >18 Å away from the reactive Asp$_{54(RR)}$, makes phosphoryl back-transfer P~DesR→DesK impossible (*Figure 4E*). Although sequence divergence makes the DHp rotational shift unlikely in HisKA HKs (*Marina et al., 2005*), DHp kinking by means of a conserved proline (*Casino et al., 2014*; *Mechaly et al., 2014*) could regulate the separation of the phosphorylatable His from the reaction center, analogously to the HisKA_3 rotational switch. Additional mechanisms that stabilize an 'open' state of HisKA HKs via extensive CA-DHp interfaces, have been reported to favor the phosphatase reaction (*Dubey et al., 2016*) further supporting our hypotheses.

This DHp helical switch is thus coupled to signal sensing, such that sensory domain rearrangements lead to intra-cytoplasmic HK coiled-coil disruption. Possibly implying trans-membrane helix tilting, or other signal-triggered helical reconfigurations, a direct connection of sensory to intracyto-plasmic α-helices can be posited. This switch can be readily understood as an order-disorder transition, which has been observed in HKs and other modular proteins, to work as an intraprotein signaling mechanism (*Schultz and Natarajan, 2013*). Helical rotational shifts have also been uncovered in structural elements within chemotaxis and TCSs, such as in HAMP domains, likely revealing common underlying transduction mechanisms eventually controlling their physiologic directionality (*Airola et al., 2013*).

HK activation leads to autophosphorylation and subsequent switching to its phosphotransferase state (*Figure 6B*). Comparing the conformation of the phosphorylatable histidine between HKs performing auto-phosphorylation (*Casino et al., 2014*; *Mechaly et al., 2014*), and the phosphotransferase complex (this study), the His residue shows ~180° flipping of its imidazole side chain (*Figure 6B*). During auto-phosphorylation, His$_{N\delta1}$ has been observed to H-bond with the ultra-conserved acidic His + 1 residue, which corresponds to Asp$_{189(HK)}$ in DesK. Such H-bond enhances His Nε2 nucleophilicity (*Quezada et al., 2005*), and indeed a carboxylate at His + 1 is critical for autophosphorylation in a number of HKs (*Willett and Kirby, 2012*). Once phosphorylated, the His disrupts its contact with this neighbor carboxylate group, stabilizing the P~HK species and leading to His $\chi-2$ dihedral angle flipping (*Figure 6C*). Subsequent re-protonation of His$_{188}$Nδ1, triggered by RR binding, is consistent with P-N bond weakening, as a first step towards phosphotransfer (*Figure 4E*). This

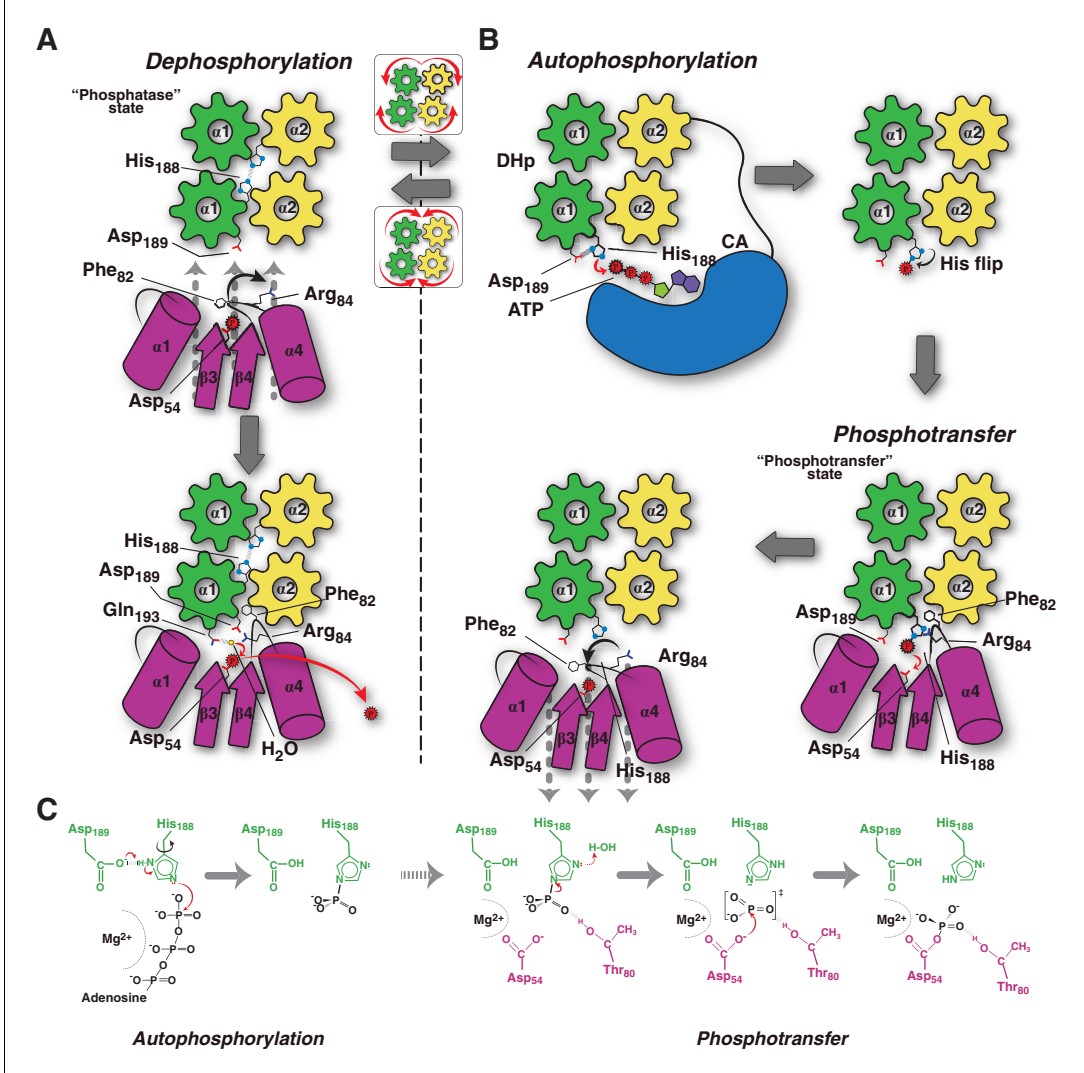

**Figure 6.** Conceptual model of TCS unidirectional signaling. (**A**) P~DesR dephosphorylation catalysis is favored by the opening of the RR phosphorylation lid, via Phe$_{82(RR)}$ insertion into an HK hydrophobic pocket, and an ionic interaction between Asp$_{189(HK)}$ and Arg$_{84(RR)}$. The Phe$_{82(RR)}$ movement allows for entry of a nucleophilic water (yellow sphere) positioned by Gln$_{193(HK)}$. In the phosphatase state the phosphorylatable His$_{188(HK)}$ is buried within the DHp domain core, avoiding autophosphorylation and phosphoryl back-transfer reactions. (**B**) Signal-mediated pathway activation triggers the cogwheel rotation of the HK DHp α-helices, via coiled-coil disruption. The His$_{188(HK)}$ becomes exposed to the solvent, CA domains are released, and autophosphorylation thus enabled. Binding to unphosphorylated RR positions the P~His$_{188(HK)}$ into a suitable orientation for phosphoryl-transfer to occur through a loose (dissociative) nucleophilic substitution. Movement of Thr$_{80(RR)}$ is coupled with phosphoryl group migration, triggering RR's phosphate lid closure and P~RR release. (**C**) Reaction scheme summarizing the autophosphorylation and phosphotransfer reactions. Stabilization of a protonated Nδ1 tautomer of the reactive histidine, is required in autophosphorylation. In keeping with the imidazole aromaticity, the nucleophilicity of Nε2 is thus finely regulated, allowing for phospho-acceptor/donor roles of the His.

mechanism might be critical in dissociative P~HK→RR reactions, which lack the simultaneous nucleophilic attack that the receiver Asp$_{(RR)}$ carries out in associative ones.

Our model predicts an important role of the highly conserved Thr$_{80(RR)}$ in stabilizing an intermediary metaphosphate group, ideally positioned to accompany the migrating phosphoryl moiety. This threonine has been substituted in CheY (*Ganguli et al., 1995*; *Appleby and Bourret, 1998*) and DosR (*Gautam et al., 2011*), suggesting it is not directly engaged in catalysis, but does play important roles in phosphoryl-transfer in vitro and proper in vivo signaling. Systematic mutagenesis studies substituting this threonine in different TCS pairs are needed to be conclusive about its role in phosphotransfer. Once the phosphotransferase reaction is complete, immediate dephosphorylation of

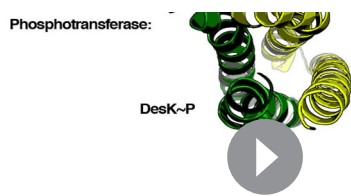

**Video 1.** Animation illustrating the phosphotransferase to phosphatase transition, within the model of TCS-mediated phosphorylation and dephosphorylation of the RR component.     As an animated support to main *Figure 6*, the fixed ground states correspond to crystal structures, and the moving intermediates in between obtained by linearly interpolated morphing, as implemented in Pymol v 1.8.2 (*Schrodinger, LLC, 2015*). This animation does not include the kinase's auto-phosphorylation step, shown at the beginning of *Figure 6B*. Structures of phosphorylated ('active') and non-phosphorylated ('inactive') DesR, when shown separate from the HK DesK, correspond respectively to PDB structures 4LE0 and 4LE1 (*Trajtenberg et al., 2014*). On the other hand, the fixed positions of P~DesK in complex with DesR (phosphotransferase state), and of DesK in complex with P~DesR (phosphatase state), correspond to the structures reported in this work (PDBs 5IUK and 5IUN, respectively). The reactive $His_{188(HK)}$ on DesK is depicted as spheres, as well as the active $Asp_{54(RR)}$ and $Phe_{82(RR)}$ on DesR. The rest of the proteins are rendered as cartoons. Chains A and B of DesK are colored in green and yellow, respectively. DesR is blue when not phosphorylated, and magenta once activated by phosphorylation. The P~His was modeled into DesK as explained in the text and Materials and methods. P~DesR corresponds to the phosphomimetic $BeF_3^-$ bonded to $Asp_{54(RR)}$.

the $P{\sim}Asp_{54(RR)}$, as well as P~DesR→DesK back-transfer, are minimized. The $\beta 4\alpha 4$ loop in DesR plays a key role in these controls, remodeling its configuration in concert with DesK-association. This loop, acting as a phosphate lid, is open in the phosphotransferase complex, enabling the transfer reaction (*Video 1*). Yet, the phosphate lid is expected to close in, following the phosphoryl migration during transfer, due to the strong phosphoryl-$Thr_{80(RR)}$ interaction. Phosphotransfer thus results in a fully activated RR configuration, with a closed phosphate lid that ensures the release of a stable P~DesR species (*Figure 6B* and *Video 1*), as observed in active free RR structures (*Trajtenberg et al., 2014*).

The $\beta 4\alpha 4$ loop amino acid sequence is not well conserved among different RRs and RR families (*Page et al., 2016*), despite its direct implication in forming the HK:RR interface. Conspicuous examples of variable residues can be mapped to $Thr_{81(RR)}$, well positioned to interact with the phosphoryl group of P~DesR, or yet $Phe_{82(RR)}$ and $Arg_{84(RR)}$, key players in the open/closure switch of DesR's phosphate lid. Some of these positions had already been pinpointed as highly variable, yet functionally relevant (*Thomas et al., 2013*; *Immormino et al., 2016*; *Page et al., 2016*), recognizing positions T + 1 and T + 2 (starting from T, the conserved position $Thr_{80(RR)}$) among others, as important modulators of RRs' catalytic activities. Particular sequence signatures of these variable motifs, have been found to correlate with RR families, in turn connected to the distinct physiological constraints of each family's signaling functions (*Page et al., 2016*). Our data now illustrate an additional selective pressure dimension to such variable positions, linked to the evolution of the HK:RR interface. Beyond sequence variability, the RR $\beta 4\alpha 4$ loop is bound to play functional roles, as it is always located at the HK:RR interface in available structures. Mutagenesis studies with functional readouts including HK-mediated activities are needed, together with a larger number of 3D structures of HK:RR complexes from different TCS families, in order to delimit universal TCS mechanisms and system-dependent variations.

## Associative vs dissociative phosphotransfer: evolution of signaling directionality

Regulating the direction of information flow is central to signaling performance. The energies of P-N and P-O bonds engaged in phosphoryl-transfer reactions among histidine and aspartate, appear to be equivalent, explaining a few known reversible systems (*Burbulys et al., 1991*). However, and except for such systems where P~Asp→His phosphorylation has been selected (such as in phosphorelays), phosphoryl back-transfer (P~RR→HK) is typically minimized. Alternative sources of reactional asymmetry, other than bonding energetics, must be at play. Taking into account that phosphotransfer is a nucleophilic substitution at phosphorus, the reaction coordinate distance (i.e. between phosphorus and the entering atom, before the transfer reaction has begun) strongly suggests that P~DesK→DesR phosphotransfer occurs through a largely dissociative, or 'loose' mechanism

(*Lassila et al., 2011*): a reaction coordinate distance of $\geq 4.9$ Å provides room for a fully dissociated metaphosphate intermediate (*Mildvan, 1997*). This distance is directly correlated with the one separating side chain N and O atoms, respectively on the reactive His and Asp residues. Considering van der Waals radii, full dissociation of the phosphate intermediate is predicted for N-O distances of $\geq 6.6$ Å. This threshold distance classifying phosphotransfer reactions into loose *vs* tight nucleophilic substitutions, nicely discriminates signaling complexes involved in, respectively, phosphorylations from the histidine imidazole group, or towards it (*Figure 5D*, *Table 6*). The His-Asp distance thus correlates with associative *vs* dissociative phosphotransfer, but also with the symmetry of the $Mg^{2+}$ cation with respect to the entering and leaving atoms (*Figure 5C and D*). Increased back-transfer has been observed when the cation is removed (*Albanesi et al., 2004*; *Lukat et al., 1990*; *Shi et al., 1999*), further supporting that $Mg^{2+}$ asymmetry favors irreversible P~His→Asp transfer reactions. Furthermore, mutations that likely increase the distance between TCS partners, without impeding their association, have been reported to affect differentially the forward- and back-transfer directions in otherwise reversible phosphoryl-transfer steps (*Janiak-Spens et al., 2005*).

The DesK:DesR structures disclose the catalytic centers for TCS-mediated dephosphorylation and phosphotransfer reactions, built with contributing residues from both partners. Moreover, the HK serves as the driving component to modulate the construction of a given active site, selecting a conformation of the RR that is prone for transfer reactions on its catalytic aspartate. Except for the phosphate lid, the REC domain bound to the HK largely resembles the RR's active form, shifting the position of the $\beta 1\alpha 1$ loop and helix $\alpha 1$, now able to recruit the $Mg^{2+}$ cation essential for phosphoryl-transfer catalysis. The phosphatase/phosphotransferase transition, switches the distance and orientation of the HK histidine side chain with respect to the RR aspartate, enabling for the distinct reaction mechanisms to take place, and ultimately ensuring the unidirectional flow of information.

## Materials and methods

### Cloning and mutagenesis

For protein crystallization purposes, recombinant protein expression plasmid pACYC-DesKC$_{STAB}$: DesR-REC was generated by sub-cloning DesKC$_{STAB}$ from pHPKS/Pxyl-desKSTA (*Saita et al., 2015*) into pACYC-DesKC$_{H188E}$:DesR-REC (*Trajtenberg et al., 2014*) using RF cloning (*Unger et al., 2010*) with primers STAB_F (5'-CCTGTATTTTCAGGGATCC GGTATTATAAAACTTCGCAAG-3') and STAB_R (5'-GTCAGACACTGTAATCACAACTTCCTTCCAG-3'). The two expression cassettes present in the pACYC-DesKC$_{STAB}$:DesR-REC encode the two recombinant proteins each fused to a His-tag and a TEV protease cleavage site.

For phosphotransfer assays, two DesR point mutants Phe82Ala and Arg84Ala were designed by mutagenesis of pQE80-DesR-REC (*Trajtenberg et al., 2014*), using mutagenic primers DesR_Fw (5'-GATTAGTATATTTATTGCAGAAGATCAGCAAATGC-3'), F82A_Rev (5'-CGGGTCTGGCGGCGGTGG TTAAG ATGATAATTTTG-3') and DesR_R84A_Rev (5'-GAAAGTAACCGGGTGCGGCGAAGGTGG TTAAG-3'), resulting in plasmids pQE80-DesR-REC$_{F82A}$ and pQE80-DesR-REC$_{R84A}$. Cloning procedures, as well as protein expression, crystallization and structure solution methods, are described in more detail at Bio-protocol (*Imelio et al., 2017*).

### Protein expression, purification and crystallization

Recombinant proteins to be used in crystallization were co-expressed from pACYC-DesKC$_{STAB}$: DesR-REC and pACYC-DesKC$_{H188E}$:DesR-REC plasmids, transformed in *Escherichia coli* strain BL21 (DE3) (Novagen). For functional assays proteins were expressed from plasmids pQE32-DesKC (*Albanesi et al., 2009*), pQE80-DesR-REC (*Trajtenberg et al., 2014*), pQE80-DesR-REC$_{F82A}$ and pQE80-DesR-REC$_{R84A}$, in *E. coli* TOP10F' (Invitrogen). Proteins were expressed and purified as previously described (*Albanesi et al., 2009*; *Trajtenberg et al., 2014*). For the DesK-DesR complexes the last size exclusion chromatography step (HiLoad 16/60 Superdex 75 preparation grade column; GE Healthcare) was used to select those fractions corresponding to the complex only.

Hanging-drop crystallizations were performed at 20°C in Linbro plates. The DesKC$_{STAB}$:DesR-REC complex (10 mg/mL protein prepared in 5 mM AMP-PCP, 10 mM MgCl$_2$) crystallized in 1 mL mother liquor [30% (w/v) PEG 4000, 0.1 M Tris.HCl pH 8.5, 0.2 M Li$_2$SO$_4$]. Protein drops were set up by mixing 0.8 µL mother liquor + 2 µL protein solution + 1.2 µL additive solution [27% (v/v) PEG 600, 0.1 M

MES pH 6.5, 0.15 M $MgSO_4$, 5% (v/v) glycerol]. Initial crystals were optimized by microseeding. Cryo-protection was achieved by slowly adding 4 µl of cryoprotection solution [32% (w/v) PEG 4000, 0.1 M Tris.HCl pH 8, 0.2 M $Li_2SO_4$, 20 mM $MgCl_2$, 18 mM $BeF_3^-$, 5 mM AMP-PCP, 15% (v/v) glycerol], then soaked in 100% cryoprotection solution and frozen in liquid $N_2$. The $DesKC_{H188E}$:DesR-REC complex (8.3 mg/mL protein prepared in 5 mM AMP-PCP, 20 mM $MgCl_2$) crystallized in mother liquor 18% (w/v) PEG 3350, 0.3 M tri-potassium citrate. Protein drops were set up by mixing 2 µL protein + 2 µL mother liquor. Cryo-protection was achieved by quick soaking in 20% (w/v) PEG 3350, 0.3 M tri-potassium citrate, 5 mM AMP-PCP, 25% (v/v) glycerol, and 20 to 150 mM $MgCl_2$ + 0 to 5 mM $BeF_3^-$. *wt* P~DesKC was crystallized as described (*Albanesi et al., 2009*), except that 5 mM ATP was used throughout instead of AMP-PCP.

## X ray diffraction data processing, crystal structure solution, refinement and analysis

Single crystal X ray diffraction was performed with a copper rotating anode home-source (Protein Crystallography Facility, Institut Pasteur de Montevideo) or synchrotron radiation (Soleil, France). Data processing was performed with autoPROC (*Vonrhein et al., 2011*). Structures were solved by molecular replacement (*McCoy et al., 2007*), using an in silico-generated model of a DesKC:DesR complex (*Trajtenberg et al., 2014*) as search probe. The other domains were then manually located in the electron density maps, and complete models were rebuilt using Coot (*Emsley et al., 2010*) and refined with Buster (*Bricogne et al., 2009*). Validation was done throughout and towards the end of refinement, using MolProbity tools (*Chen et al., 2010*). DesKC:DesR-REC surface complementarity was calculated according to (*Lawrence and Colman, 1993*), resulting in figures of ~0.6. Visualization of protein models and structural analyses, figure rendering and morphing for animation were performed with Pymol (*Schrodinger, LLC, 2015*). Software for data processing, structure determination and analysis was provided by the SBGrid Consortium (*Morin et al., 2013*).

## Small angle X-ray scattering data acquisition and analysis

SEC-SAXS experiments were carried out at beamline SWING (Soleil synchrotron, France). Purified DesK-DesR complexes samples were brought to 10 mg/mL in 50 mM Tris.HCl pH 8.0, 300 mM NaCl, and injected into a Superdex 75 5/150 GL column (GE Healthcare Biosciences) equilibrated in the same buffer. Samples were eluted at constant flow (0.15 mL/min) and loaded into capillary cell for X-ray exposure. Recorded frames were processed with Foxtrot (*David and Pérez, 2009*) following standard procedures. Subsequent analyses of the scattering data were performed with tools from the ATSAS (*Konarev et al., 2006*). Forward scattering $I_{(0)}$ and the particle's radius of gyration ($R_g$) were estimated using the Guinier approximation. $I_{(0)}$ real was calculated as the extrapolated intensity at zero scattering angle. Pair-distance distribution functions were calculated from the scattering patterns with GNOM, which also provides maximum particle dimension ($D_{max}$) and $R_{g\ (real)}$ values. Molecular mass (MM) of particles was estimated from the volume-of-correlation ($V_c$) values (*Rambo and Tainer, 2013*). Theoretical scattering patterns from atomic models were calculated and fitted to experimental curves using CRYSOL. Mixture analysis was performed with OLIGOMER.

## Isothermal titration calorimetry

Isothermal titration calorimetry (ITC) assays were performed on a VP-ITC (MicroCal Inc., Northampton, MA). Titrations consisted of an initial injection (1 µL), followed by 25–30 injections of 10 µL of DesR-REC (ligand) on the cell containing the different DesKC variants (partner). Assays were carried out at 15°C in a buffer containing 20 mM Tris.HCl pH 8, 0.3 M NaCl, 10 mM $MgCl_2$ and 0.5 mM AMP-PNP. The heat released by the dilution of the ligand was determined injecting the ligand, on the cell containing the working solution without partner, using the same sequence of injections. The concentrations used for the experiments were between 25–30 µM for DesKC and 350–400 µM of DesR-REC and each titration was done in duplicate. The data were analyzed with MicroCal Origin version 7 software (MicroCal Software Inc.), after manual baseline correction, and subtraction of heat due to ligand dilution. Binding isotherms were fitted to a two independent sequential site model.

## Sequence-based direct coupling analysis of the HK:RR complex

Sequences of HK and RR pairs, with the same architecture as DesK and DesR (families HisKA_3 and NarL, respectively), belonging to the same operon, were selected from the Uniprot database using hmmsearch (*Eddy, 1998*). Hidden Markov model-profiles for each domain were obtained from Pfam (*Finn et al., 2016*). Redundancy was filtered using a cutoff of 90% (*Li and Godzik, 2006*) resulting in a total of 3318 sequences including HKs (DHp+CA) and RRs (REC+HTH). The concatenated HK:RR sequences were aligned with hmmalign (*Eddy, 1998*) and manually curated, removing sequences with large insertions. DI (direct information) of each pair of residues of the alignment was calculated using mfDCA matlab script (*Morcos et al., 2011*) and we only take in consideration coevolving pairs of residues farther than five positions apart in sequence. The same procedure was followed for the HisKA and PhoB families for comparison purposes. A similar trend in equivalent 3D positions for covariant residue pairs among different HK and RR families was thus confirmed (*Podgornaia et al., 2013*; *Skerker et al., 2008*; *Weigt et al., 2009*).

## In silico modeling and molecular dynamics simulations

The phosphotransferase complex was built by replacing the DesKC$_{H188E}$ dimer of 5IUK with *wt* P~DesKC (PDB 5IUM). This replacement resulted in model with barely no clashes, except for P~His$_{188(HK)}$ slightly bumping into Phe$_{82(RR)}$ and Arg$_{84(RR)}$. Energy minimization was then performed on such constructed complex with all-atom constraints in Rosetta (*Das and Baker, 2008*), strongly restraining shifts to maintain the experimental coordinates. A final *wt* P~DesKC:DesR-REC model was thus obtained, with optimal stereochemical geometry and no clashes (0.09 rmsd between the energy minimized model and the pdb 5IUM experimental structure, aligning all 1930 DesK atoms). To further evaluate the minimized *wt* P~DesKC:DesR-REC model (see *Figure 4A*), molecular dynamics simulations were performed, observing that its conformation is stable throughout the trajectory (see *Figure 4B*), especially considering the positions of His$_{188(HK)}$, Asp$_{54(RR)}$ and Thr$_{80(RR)}$ in contact with the phosphate.

For molecular dynamics (MD) calculations, the missing ATP lid loop within the CA domain in chain A (residues 328 to 335) were reconstructed by using a previously reported high resolution structure (PDB 3EHG) as template. The other CA domain (in chain B, residues 241–367) was deleted. The missing N-terminal portion of chain B helix α1, was reconstructed and extended up to residue Lys$_{155(HK)}$ using 5IUK as a template.

For computational efficiency only DesK residues Lys$_{155(HK)}$ to Asn$_{368(HK)}$ in chain A and Lys$_{155(HK)}$ to Ser$_{239(HK)}$ in chain B were considered. Only His$_{188(HK)}$ in chain B was set to be phosphorylated. His$_{335(HK)}$ was protonated at position Nδ1, while other histidines were protonated at Nε2, to preserve the interaction network. The ATP moiety and Mg$^{2+}$ cation present within the CA domain were both kept in the model. The structure of DesR-REC spanned residues Ser$_{0(RR)}$ to Leu$_{131(RR)}$. The Mg$^{2+}$ cation in the active site of DesR-REC and the three crystallographic water molecules of the metal's coordination sphere, were included in the model.

Acetyl and N-methylamide capping groups were added to N- and C- terminal residues, which did not correspond to the real terminals in the full protein sequences. The protein complex was solvated within an octahedral box of 15 Å from the solute, and sodium counterions were added to preserve the electroneutrality of the system. The AMBER force field ff14SB (*Maier et al., 2015*) was used to represent the amino acids. The parameters for His$_{188(HK)}$ phosphorylated at NE2 with an unprotonated phosphate group were taken from (*Homeyer et al., 2006*). The ATP molecule was described with reported parameters (*Meagher et al., 2003*). The TIP3P model (*Jorgensen et al., 1983*) was used for water molecules. Monovalent ions were treated with previously described parameters (*Joung and Cheatham, 2008*), while Mg$^{2+}$ ions were modeled using the compromise set of parameters for the 12–6 non-bonded potential in TIP3P (*Li et al., 2013*).

All calculations were performed with the GPU version of AMBER14 (*Salomon-Ferrer et al., 2013*). Initially, the whole system was relaxed by energy minimization, then it was equilibrated for 0.2 ns in the NVT ensemble imposing harmonic positional restrains of 10 kcal mol$^{-1}$ Å$^{-2}$ on the protein atoms. A reference temperature of 300 K was set by coupling the system to the Langevin thermostat (*Pastor et al., 1988*; *Wu and Brooks, 2003*) with a friction constant of 50 ps$^{-1}$, which approximates the physical collision frequency for liquid water (*Izaguirre et al., 2001*). A 10 Å cut-off was used for non-bonded interactions, while long-range electrostatics were evaluated using Particle

Mesh Ewald (PME) (*Darden et al., 1993*; *Essmann et al., 1995*). A time step of 2 fs was used and all bonds involving hydrogen atoms were restrained using the SHAKE algorithm (*Miyamoto and Kollman, 1992*; *Ryckaert et al., 1977*). Production simulations were performed in the NPT ensemble. The pressure was kept at 1 atm by means of the Berendsen barostat (*Berendsen et al., 1984*). Snapshots were recorded every 5 ps for analysis.

## Autodephosphorylation and phosphotransfer assays

To purify the phosphorylated species of DesR-REC and DesR-REC$_{F82A}$, 600 μM of both recombinant proteins were auto-phosphorylated using 50 mM acetyl phosphate in a buffer containing 25 mM Tris.HCl pH 8, 300 mM NaCl and 30 mM MgCl$_2$, at room temperature. Reactions were stopped by adding EDTA to a final concentration of 50 mM, and buffer exchanged by using a PD Minitrap G-25 (GE Healthcare) desalting column. Auto-dephosphorylation assays of P~DesR-REC variants were performed at 30 μM of protein concentration and incubated in the presence of 30 mM MgCl$_2$. At different time points the reactions were stopped by adding SDS-PAGE sample buffer. For each time point DTT was added to a final concentration of 25 mM and incubated for 5 min. DTT in excess was blocked with 40 mM iodo-acetamide and loaded in a Phos-tag acrylamide SDS-PAGE, as described before (*Trajtenberg et al., 2014*), Coomassie blue-stained gels were scanned and quantification of all reactions was done by densitometry using ImageJ (*Schneider et al., 2012*).

To perform phosphotransfer assays (*Figure 4E*), DesKC was phosphorylated by incubation with 10 mM ATP and 20 mM MgCl$_2$ at 100 μM of protein, for 1 hr at room temperature. P~DesKC was then purified by size exclusion chromatography S75 10/300 (GE Healthcare), equilibrated in 20 mM Tris.HCl pH 8, 300 mM NaCl. Phosphotransfer reactions were performed in triplicate by combining 30 μM P~DesKC with equimolar concentrations of DesR-REC (both wild type and DesR-REC$_{F82A}$), at 25°C in 20 mM Tris.HCl pH 8, 300 mM NaCl, 30 mM MgCl$_2$. Reactions were stopped at different time points as described above. Samples were then separated by Phos-tag SDS-PAGE.

In order to assess unidirectionality (*Figure 5A*), the reverse and forward reactions were analyzed. The forward reaction was prepared as described above using P~DesKC as phosphodonor for 1 min. On the other hand, the reverse of the reaction was measured by incubating 30 μM of P~DesR-REC (wild type or point mutants DesR-REC$_{F82A}$ or DesR-REC$_{R84A}$) with 30 μM DesKC, 50 mM acetyl phosphate and 30 mM MgCl$_2$, for 1 min at room temperature. Reactions were stopped as described above and samples analyzed by Phos-tag SDS-PAGE and quantified by densitometry.

## Accession numbers

The X ray structures presented have been deposited in the wwPDB with accession codes 5IUJ (DesK-DesR complex in the phosphotransfer state with low Mg$^{2+}$ [20 mM]), 5IUK (DesK-DesR complex in the phosphotransfer state with high Mg$^{2+}$ [150 mM]), 5IUL (DesK-DesR complex in the phosphotransfer state with high Mg$^{2+}$ [150 mM] and BeF$_3^-$), 5IUM (phosphorylated wild type DesKC) and 5IUN (DesK-DesR complex in the phosphatase state).

Raw X ray diffraction data corresponding to each one of these structures are publicly available at SBGrid Data Bank (http://data.sbgrid.org) as dataset entries 399, 400, 401, 407 and 408.

## Acknowledgements

We want to thank the staffs at synchrotron beamlines Proxima 1 and 2, Soleil (France), especially William Shepard; and Daniela Albanesi for kindly providing plasmid pHPKS/Pxyl-desKSTA. The Institut Pasteur International Network is acknowledged for support through the IMiZA International Joint Unit. We acknowledge funding from Agencia Nacional de Investigación e Innovación (ANII), Uruguay (FCE2009_1_2679; FCE2007_219); Agence Nationale de la Recherche (ANR), France (PCV06_138918); Centro de Biología Estructural del Mercosur (www.cebem-lat.org) and Fondo para la Convergencia Estructural del MERCOSUR (COF 03/11).

## Additional information

### Funding

| Funder | Grant reference number | Author |
|---|---|---|
| Agencia Nacional de Investi-gación e Innovación | FCE2009_1_2679 | Felipe Trajtenberg<br>Alejandro Buschiazzo |
| Agence Nationale de la Re-cherche | PCV06_138918 | Alejandro Buschiazzo |
| FOCEM (MERCOSUR Structur-al Convergence Fund) | COF 03/11 | Alejandro Buschiazzo |
| Centro de Biologia Estructural del Mercosur | | Alejandro Buschiazzo |
| Agencia Nacional de Investi-gación e Innovación | FCE2007_219 | Felipe Trajtenberg<br>Alejandro Buschiazzo |

The funders had no role in study design, data collection and interpretation, or the decision to submit the work for publication.

### Author contributions

FT, AB, Conception and design, Acquisition of data, Analysis and interpretation of data, Drafting or revising the article; JAI, GO, Acquisition of data, Analysis and interpretation of data; MRM, Acquisition of data, Analysis and interpretation of data, Contributed unpublished essential data or reagents; NL, Acquisition of data, Contributed unpublished essential data or reagents; MAM, Analysis and interpretation of data, Drafting or revising the article, Contributed unpublished essential data or reagents; AEM, Acquisition of data, Analysis and interpretation of data, Drafting or revising the article

### Author ORCIDs

Alejandro Buschiazzo, http://orcid.org/0000-0002-2509-6526

## Additional files

### Major datasets

The following datasets were generated:

| Author(s) | Year | Dataset title | Dataset URL | Database, license, and accessibility information |
|---|---|---|---|---|
| Trajtenberg F, Buschiazzo A | 2016 | Crystal structure of phosphorylated DesKC | http://www.rcsb.org/pdb/search/structid-Search.do?structureId=5IUM | Publicly available at the RCSB Protein Data Bank (accession no: 5IUM) |
| Trajtenberg F, Imelio JA, Larrieux N, Buschiazzo A | 2016 | Crystal structure of the DesK-DesR complex in the phosphatase state | http://www.rcsb.org/pdb/search/structid-Search.do?structureId=5IUN | Publicly available at the RCSB Protein Data Bank (accession no: 5IUN) |
| Trajtenberg F, Imelio JA, Larrieux N, Buschiazzo A | 2016 | Crystal structure of the DesK-DesR complex in the phosphotransfer state with low Mg2+ (20 mM) | http://www.rcsb.org/pdb/search/structid-Search.do?structureId=5IUJ | Publicly available at the RCSB Protein Data Bank (accession no: 5IUJ) |
| Trajtenberg F, Imelio JA, Larrieux N, Buschiazzo A | 2016 | Crystal structure of the DesK-DesR complex in the phosphotransfer state with high Mg2+ (150 mM) | http://www.rcsb.org/pdb/search/structid-Search.do?structureId=5IUK | Publicly available at the RCSB Protein Data Bank (accession no: 5IUK) |
| Trajtenberg F, Imelio JA, Larrieux N, Buschiazzo A | 2016 | Crystal structure of the DesK-DesR complex in the phosphotransfer state with high Mg2+ (150 mM) and BeF3 | http://www.rcsb.org/pdb/search/structid-Search.do?structureId=5IUL | Publicly available at the RCSB Protein Data Bank (accession no: 5IUL) |
| Trajtenberg F, | 2016 | Complex DesKC:DesR-REC (B. | http://dx.doi.org/10. | Publicly available at |

| | | | | |
|---|---|---|---|---|
| Buschiazzo A | | subtilis) - phosphotransferase state, low Mg2+ | 15785/SBGRID/399 | the SBGrid database (dataset no. 399; X-ray diffraction raw data for PDB 5IUJ) |
| Trajtenberg F, Buschiazzo A | 2016 | Complex DesKC:DesR-REC (B. subtilis) - phosphatase state | http://dx.doi.org/10.15785/SBGRID/400 | Publicly available at the SBGrid database (dataset no. 400; X-ray diffraction raw data for PDB 5IUN) |
| Trajtenberg F, Buschiazzo A | 2016 | Complex DesKC:DesR-REC (B. subtilis) - phosphotransferase state, high Mg2+ | http://dx.doi.org/10.15785/SBGRID/401 | Publicly available at the SBGrid database (dataset no. 401; X-ray diffraction raw data for PDB 5IUK) |
| Buschiazzo A | 2016 | Phosphorylated DesKC (B. subtilis) | http://dx.doi.org/10.15785/SBGRID/407 | Publicly available at the SBGrid database (dataset no. 407; X-ray diffraction raw data for PDB 5IUM) |
| Trajtenberg F, Buschiazzo A | 2016 | Complex DesKC:DesR-REC (B. subtilis) - phosphotransferase state, high Mg2+ and BeF3- | http://dx.doi.org/10.15785/SBGRID/408 | Publicly available at the SBGrid database (dataset no. 408; X-ray diffraction raw data for PDB 5IUL) |

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
