## [Decision Letter]

Thank you for submitting your article "Regulation of signaling directionality revealed by 3D snapshots of a kinase:regulator complex in action" for consideration by *eLife*. Your article has been favorably evaluated by Michael Marletta (Senior editor) and three reviewers, one of whom, Michael Laub, is a member of our Board of Reviewing Editors. The following individuals involved in review of your submission have agreed to reveal their identity: Robert Bourret (Reviewer #2).

The reviewers have discussed the reviews with one another and the Reviewing Editor has drafted this decision to help you prepare a revised submission.

Summary:

Two-component regulatory systems, comprised of histidine kinases and response regulators, are a widespread means of signal transduction in bacteria, and are also found in archaea, eukaryotic microorganisms, and plants. Many, if not most, histidine kinases are bifunctional, capable of transferring phosphoryl groups to their cognate response regulator and stimulating their dephosphorylation. There are many unanswered questions about the mechanisms of the phosphotransfer and phosphatase reactions, in part because of the paucity of high-resolution structures of HK•RR complexes. The manuscript by Tratjenberg et al. reports multiple DesK•DesR structures and the reviewers all agreed that it will likely be regarded as a landmark study. In general, the manuscript is well written and illustrated, providing important new insights into (i) the mechanism for switching the HK between phosphotransfer and phosphatase reactions, (ii) a novel conformation of the RR when complexed with the HK, (iii) key roles of variable RR residues in the reactions, (iv) a dissociative phosphotransfer reaction mechanism, and (v) the apparent lack of participation of the conserved His on the HK in the phosphatase reaction.

There were, however, some concerns about how the data were presented and, in some cases, interpreted.

Essential revisions:

1) The Results section begins with two key conclusions, that "Two crystal structures of the DesKC:DesR complex represent snapshots of the dephosphorylation and the phosphotransfer reactions". The authors eventually make a reasonable case that the structures are what they are interpreted to be. However, the text first asserts that the structures represent the two reactions and then discusses some properties of each structure before explaining why the authors believe the structures represent the two reactions. It would be much more convincing to the reader to first provide the reasons why the authors believe their structures represent the phosphotransfer and dephosphorylation reactions and then describe the structures.

2) The authors gloss over some underlying weakness in their claims about what the structures represent. For example, the authors claim that DesKCSTAB is a phosphatase-constitutive mutant. This is true in the sense that the phosphatase activity of DesKCSTAB is (i) not regulated by temperature and (ii) comparable to the high temperature value exhibited by wild type DesKC. However, the actual phosphatase activity reported in Figure 4 of Saita et al. 2015 is quite weak. When mixed with DesR-P in equimolar amounts, DesKCSTAB was able to dephosphorylate only half of the DesR-P, and did so at twice the initial rate of wild type DesKC. In other words, there is no evidence that one DesKCSTAB can act as an enzyme to dephosphorylate more than one DesR-P molecule.

3a) Figure 3 and the text in subsection “A single response regulator conformation is selected by the two functional states of the kinase” compare "inactive" and "active" structures of DesR REC with the conformation of DesR REC observed in complexes. It is worth noting that the structure used to represent the inactive state (4LE1) lacks not only a phosphoryl group analogue, but also the divalent metal ion necessary for phosphoryl group chemistry. There are some differences in structures of receiver domains with or without metal ion bound, so the lack of a metal ion probably exaggerates the differences between the "inactive" state and the conformation observed in the complexes. It is perfectly appropriate to use 4LE1 when a metal bound structure is not available; however, it would be better to clearly alert the reader to the lack of metal ion.

3b) In the structure showed for the phosphotransferase and phosphatase reactions, the RR configuration is similar, corresponding in both cases to a partially activated (phosphorylated) conformation. However, in the phosphotransferase reaction the apo-RR should be recognized by P-HK, while the P-RR is recognized in the phosphatase reaction by the apo-HK. As is well know and the authors mentioned, the apo-RR and P-RR conformations are different. Therefore, the authors should analyze and discuss how are these conformations selectively recognized by the complementary HK conformation (structural determinants).

4) The authors missed some opportunities for more effective communication of key results. In particular, the mechanism of phosphatase activity by HK remains unresolved and whether or not the conserved His residue participates in the reaction is controversial. For example, a recent prominent review article [Bhate, Structure 23, 981 (2015)] states that although the His is not always required, "in cases where the catalytic histidine is required, it likely acts as a base to assist the attack of water or hydroxide on the phosphoryl-aspartyl group". The findings reported in this manuscript place the His far from the attacking water.

5) Similarly, for readers interested in mechanistic details, it would be helpful to include illustrations of the proposed active sites for the phosphotransfer and phosphatase reactions showing just the relevant side chains and small molecules, and indicating distances and geometries, without the clutter of backbone ribbons.

6) There is great merit in providing a proposed mechanism such as in Figure 6, which provides specific features that can be tested. I have two comments about the illustration and accompanying text:a) In subsection “A HK:RR concerted switch controls efficiency along the signaling pathway” the authors say that a hydrogen bond is formed between His188 and Asp189. Is this modeled in the reported structures or just speculation? The reference cited (Quezada et al. 2005) actually demonstrated hydrogen bonding between the conserved His and a Glu that is 22 residues to the C-terminal side, not 1 residue C-terminal as in DesK. It might be worth citing Willett & Kirby, PLOS Genetics 8, e1-003984 (2012), which provides evidence in multiple HKs that the acidic residue at variable position H^+^1 is critical for autophosphorylation, but not for phosphatase activity.b) The proposed dissociative phosphotransfer mechanism involves a key interaction between the conserved Ser/Thr residue of the RR and the phosphoryl group that has been released from the HK, before attack by the RR Asp. One prediction of this mechanism might be that phosphotransfer from HK to RR would be severely affected in the absence of the conserved Ser/Thr residue. I am aware of only one potentially relevant piece of data in the literature. Figure 3 of Appleby & Bourret, J Bacteriol 180, 3563 (1998) shows that phosphotransfer occurs from CheA to a CheY mutant in which the conserved Thr is repaced by an Ala. Obvious caveats are that CheAs are different than other HKs, and that the experiment was not particularly sensitive to changes in phosphotransfer rate (i.e. the rate of phosphotransfer could have been dramatically slower but still observed).

7) The apparently important role played in the reactions by variable RR residues is very interesting. For example, subsection “The phosphatase to phosphotransferase transition: reconfiguring the reaction center for catalysis” notes apparent hydrogen bonding between the phosphoryl group and Thr81 of DesK. One wonders what happens in the ~80% of RRs for which the residue corresponding to Thr81 is not a Ser or Thr. Similarly, in this section and elsewhere, the role of Phe82 in forming the phosphate lid is noted. One wonders what happens in the 2/3 of response regulators that do not contain an aromatic residue (Tyr, Phe, His) at the position corresponding to Phe82. These are rich topics for future investigation. It might be worth noting someplace in the manuscript that the reported structures suggest important roles for variable residues, which by definition vary between systems.

---

## [Author Response]

*Essential revisions:*

*1) The Results section begins with two key conclusions, that "Two crystal structures of the DesKC:DesR complex represent snapshots of the dephosphorylation and the phosphotransfer reactions". The authors eventually make a reasonable case that the structures are what they are interpreted to be. However, the text first asserts that the structures represent the two reactions and then discusses some properties of each structure before explaining why the authors believe the structures represent the two reactions. It would be much more convincing to the reader to first provide the reasons why the authors believe their structures represent the phosphotransfer and dephosphorylation reactions and then describe the structures.*

We are pleased to learn that the reviewers agree with our interpretation about the structures’ biochemical meaning. We’re sorry for not being clear enough in the way the subsection was written. We have now rephrased it according to your suggestions.

Before summarizing the revisions we have introduced, we feel it is probably helpful to argue about the logic behind our original version.

i) The two key conclusions that started the section, as the editors point out, were in fact the very title of the subsection itself. So, it was our deliberate intention to summarize the main conclusions there, in an attempt to quickly help the reader grasping the important messages. Otherwise, the risk is to lose the reader’s attention along the arduous, although essential, structural descriptions.

ii) Those descriptions, based on crystallographic as well as also in solution biophysics data, were elaborated in such a way as to build the several pieces of evidence, ultimately supporting our functional interpretation for each structure. In other words, it seems like a challenging exercise to first convince the reader of our interpretation of the data, before actually introducing the data themselves.

In any case we have tried to meet this challenge in this revised version, so that the subsection reads more clearly. We have substantially rephrased the entire subsection. We prefer keeping the same title though for the reasons described above. After the title, we now continue by first stating the proposed hypotheses (about each structure’s functional meaning), to then more clearly unfold, in a sequential manner, the pieces of evidence that support our claims. We also take advantage, within this subsection, to further elaborate the arguments supporting the interpretation of the DesKC_STAB_:DesR-REC complex as a snapshot of the phosphatase reaction. This is described in the following point #2.

*2) The authors gloss over some underlying weakness in their claims about what the structures represent. For example, the authors claim that DesKCSTAB is a phosphatase-constitutive mutant. This is true in the sense that the phosphatase activity of DesKCSTAB is (i) not regulated by temperature and (ii) comparable to the high temperature value exhibited by wild type DesKC. However, the actual phosphatase activity reported in Figure 4 of Saita et al. 2015 is quite weak. When mixed with DesR-P in equimolar amounts, DesKCSTAB was able to dephosphorylate only half of the DesR-P, and did so at twice the initial rate of wild type DesKC. In other words, there is no evidence that one DesKCSTAB can act as an enzyme to dephosphorylate more than one DesR-P molecule.*

We agree with the reviewers that this may be a very important analysis to interpret Figure 4 of Saita et al., 2015 more fully, and for that matter, to extend our understanding of the HK-mediated P~RR dephosphorylation reaction.

But, within the scope of this manuscript, that DesKC_STAB_ works in comparable ways to *wt*DesKC, as if it were trapped in the *wt* phosphatase-competent state, is absolutely relevant. And we understand that this comparison indeed allows for the point to be made, as the reviewers point out in the first part of their statement. Strengthening this argument even further, it seems pertinent to analyze the data reported in Figure 2 and Figure 3 in that article (besides Figure 4, which only concerns P~RR dephosphorylation), also revealing that DesK_STAB_ is not able to turn on the signaling pathway in vivo, nor to auto-phosphorylate in vitro, in the low-temperature regime; again, indistinguishable from *wt*DesK in its phosphatase state.

Furthermore, the DesK_DEST_ mutant, which comprises amino acid substitutions in the same coiled-coil region as DesK_STAB_, but seeking for an opposite, coiled-coil-destabilizing effect, indeed results in an in vivoopposite effect, displaying a gain-of-function, constitutive ‘on’ state of the signaling pathway, irrespective of the signal (Figure 2 of Saita et al., 2015). This is also relevant, because it validates the important assumption that mutations introduced at the chosen coiled-coil region, far from the reactional active site, do not interfere with the reaction itself (in contrast to previous mutated DesKC versions trapping the phosphatase state, such as DesKC_His188Val_, where the very phosphorylatable histidine was substituted, admitting doubts about its potential effect on the reaction). Saita et al. 2015 from a functional perspective, and now this report with high resolution structural data, actually confirm that DesKC_His188Val_ was a pertinent mutant to trap a constitutive phosphatase-competent form of the HK.

DesKC_His188Val_ was originally shown to maintain P~DesR phosphatase activity both in vivoand in vitro(Albanesi et al. 2004), and its structure as a free protein (i.e. not in complex with DesR), was solved in two different crystal forms (PDBs 3EHH and 3EHJ) (Albanesi et al., 2009). Those two structures are identical to the level of precision of the data, and we now observe that they both match the kinase’s conformation of the DesK_STAB_:DesR-REC complex that we are now reporting. This is yet a further strong argument consistent with the interpretation of DesK_STAB_:DesR-REC representing the pre-dephosphorylation ground-state of the phosphatase reaction.

To highlight all these arguments, we have now further elaborated the paragraph according to previous comment #1; adding the quantitative comparison of the kinase partner from the free DesKC_His188Val_ crystal forms superposed onto the phosphatase complex we are now reporting; and included an additional panel in Figure 1—figure supplement 1 (new panel E) with these superpositions.

*3a) Figure 3 and the text in subsection “A single response regulator conformation is selected by the two functional states of the kinase” compare "inactive" and "active" structures of DesR REC with the conformation of DesR REC observed in complexes. It is worth noting that the structure used to represent the inactive state (4LE1) lacks not only a phosphoryl group analogue, but also the divalent metal ion necessary for phosphoryl group chemistry. There are some differences in structures of receiver domains with or without metal ion bound, so the lack of a metal ion probably exaggerates the differences between the "inactive" state and the conformation observed in the complexes. It is perfectly appropriate to use 4LE1 when a metal bound structure is not available; however, it would be better to clearly alert the reader to the lack of metal ion.*

We have now included the fact that 4LE1 (our “inactive” RR form to compare with) has no metal bound in the active site, both in the main text as in Figure 3’s legend.

Although beyond the scope of this manuscript, and also for the sake of limiting its length to reasonable extents, we wish to follow up on this interesting observation in the context of this exchange with the reviewers.

We believe that inactive metal-free forms should not be excluded as relevant species in vivo. Relatively scarce data are available as to the actual Mg^2+^-binding affinities of different RR REC domains (even less comparing phosphorylated vs non-phosphorylated).

One of the best studied examples in this respect, is that of CheY. This monodomain RR is claimed to be significantly metallated, even in the unphosphorylated form (so that the physiologically relevant “inactive” form of CheY would be “Mg^2+^-bound unphosphorylated” CheY). Such a claim is supported mostly on two pieces of evidence: i-in vitromeasurements of CheY’s Mg^2+^-binding affinity at neutral pH (Lukat el al 1990), which falls approximately within the same order of magnitude, sub-mM/low mM range, as that of free intra-cellular Mg^2+^ concentrations in enterobacteria (Froschauer et al. 2004 FEMS Microbiol Lett 237:49; Alattosava et al. 1985 J Bac 162:413), and ii- “direct” estimations of Mg^2+^-bound CheY by *in cell* NMR methods (Hubbard et al. 2003 Mol Microbiol 49:1191), revealing signal peaks more similar to metallated than to non-metallated CheY species. However, it must be stressed, as the authors recognize in that Hubbard et al. paper, that these *in cell* NMR measurements were performed using cells that had overexpressed CheY to reach NMR-detection limits (~5 times higher concentrations of CheY than normal), which given a constant metal-binding affinity, and a tightly regulated intra-cellular Mg^2+^ concentration, is expected to result in a ~5-fold overestimation of the complexed form (not to account for the undetermined proportion of phosphorylated vs unphosphorylated CheY variants under such overexpression regimes). Taking all this evidence together, it doesn’t really seem sensible at this time to rule out the possibility of having significant amounts of Mg^2+^-free CheY in *E. coli* cells, mostly so when cells have low free metal concentrations (1mM or lower).

A scenario where the majority of inactive RR population is expected to be metal-free, is even more likely for other RRs displaying even lower Mg^2+^-binding affinity constants, as high as ~50mM K_D_ (Guillet et al. 2002 JBC 277:42003; Bourret 2010 Curr Opin Microbiol 13:142). We have not measured the metal-binding constants for DesR REC. But, in any case, it seems to be important to use metal-free structures of RRs as inactive models whenever possible, in as much as metal-bound ones.

Our data support the hypothesis whereby the cognate kinase stabilizes (selects) a conformation of the REC domain, which is closer to the fully active form even when not yet phosphorylated nor metallated (as we had previously hypothesized only on the basis of in vivofunctional data and *in silico* modeling of the molecular details, Trajtenberg et al., 2014; now we can at last see the molecules). The series of incubations with increasing concentrations of Mg^2+^ (in the presence or absence of added BeF_3_^-^) that we now report using the phosphotransferase complex crystals, allowed us to solve three independent crystal structures, that show that Mg^2+^ binds into DesR’s active site only at higher soaked metal concentrations, with BeF_3_^-^ never being detectable: this is, we believe, a first evidence that Mg^2+^ is recruited with lower affinity due to kinase-binding mediated “preactivation”, reaching full metal occupancy in concert with actual phosphoryltransfer (implying that phosphorylation increases Mg^2+^-binding affinity, and vice-versa).

*3b) In the structure showed for the phosphotransferase and phosphatase reactions, the RR configuration is similar, corresponding in both cases to a partially activated (phosphorylated) conformation. However, in the phosphotransferase reaction the apo-RR should be recognized by P-HK, while the P-RR is recognized in the phosphatase reaction by the apo-HK. As is well know and the authors mentioned, the apo-RR and P-RR conformations are different. Therefore, the authors should analyze and discuss how are these conformations selectively recognized by the complementary HK conformation (structural determinants).*

We greatly appreciate this comment, prompting us to include a new subsection within the Discussion.

In order to limit the extension of the main text, we wish to further exchange some ideas and data within this direct discussion with the reviewers. Hopefully you will find the most important points properly summarized within the Discussion subsection that we have added in the manuscript.

Our data indeed indicate that HKs are largely unable to discriminate binding to RR or P~RR, on the sole basis of structural determinants that would stand out from comparing the two complexes.

Technical difficulties in measuring HK:RR binding affinities for specific functional states (i.e. the HK in its phosphatase state with RR *vs* P~RR, compared to the phosphotransfer-competent HK with P~RR *vs* RR), have likely hampered in-depth analyses of this important issue for many HK:RR systems. Working with *wt* proteins most often implies simultaneous phosphoryl-transfer and dephosphorylation, confusing clear-cut interpretations. With these caveats considered, only a few HK:RR interactions have been thoroughly characterized, which actually lend further support to our hypothesis. EnvZ binds OmpR and P~OmpR with equivalent affinities (Yoshida et al. 2002). The same holds for PhoQ binding PhoP or P~PhoP (Castelli et al., 2003). CheA has slightly lower affinity (but within the same order of magnitude) for P~CheY than for CheY (Schuster et al., 1993Nature 365:343; Li et al. 1995 Biochemistry 34:14626). And, this 2- to 3-fold difference cannot be immediately interpreted in terms of potential CheA:CheY interface differences, to be directly compared to our data. Most importantly because, besides CheA’s P1 domain (containing the phosphorylatable His, hence certainly contacting CheY), CheA bears an additional P2 domain, which is a CheY-binding element not present in most HKs including DesK. Given that CheA-P2 binds to a surface on CheY [Welch et al. 1998 Nat Struct Biol 5:25] that is known to change according to CheY’s phosphorylation state (the α4β5α5 surface, which is the same surface that later binds FliM during downstream output response), it would not be surprising that CheA:CheY *vs* CheA:P~CheYbinding affinities change driven by CheA-P2:CheY binding modulation. Added to this, we still don’t know whether P~CheA binds to CheY with a significantly different affinity as apo-CheA does (our data predict those association constants to be similar, further studies needed to prove it).

DesK:DesR constitutes a particularly nice model in this respect, due to our ability to trap functional states using specific point mutations, ultimately facilitating the interpretation of HK:RR binding phenomena.

As we now discuss within the manuscript, the reported crystallographic and calorimetric evidence are very much consistent, both supporting the notion that no structural determinants are relevant for the HK to choose between the apo and phosphorylated forms of its RR partner at the molecular level.

A legitimate question can immediately be asked: if there are no structure-based discriminants, how does the system work in the cell, avoiding seemingly futile associations?

More data needs to be generated in order to unveil such in vivomechanisms at the cell level, beyond the scope of this paper. We are tempted to speculate that when the signaling pathway turns the HK off, at first large amounts of P~RR are present, such that the HK in its phosphatase state catalyzes P~RR dephosphorylation. At some point though, enough amount of apo-RR builds, competing with P~RR for HK binding. This could result in a biphasic HK-mediated P~RR dephosphorylation slope (an example of product inhibition), and the RR’s autodephosphorylation capacity could become increasingly relevant (as well as additional dedicated phosphatases such as CheZ, RapH, etc, if present). Such biphasic behavior might be a better interpretation of Figure 4 of Saita et al., 2015 Mol Microbiol 98:258 (already mentioned above in response to comment #2). A further derivation of this is that fairly stable complexes of HK:RR (with neither partner phosphorylated) cannot be excluded, perhaps playing biologically relevant roles. The low nanomolar affinity (~30nM) measured for CheA:CheY supports this notion (Schuster et al., 1993Nature 365:343), as the authors explicitly mention.

In turn, when the HK is activated, its own autophosphorylation dominates at high ATP concentrations, with intramolecular CA domains outcompeting RR binding (P~RR concentrations expected to be low). P~HK will most likely interact only with apo-RR, due to the strong electrostatic repulsion force that would result of placing two phosphoryl groups close in space.

Finally, we should also emphasize that for a correct activation and inactivation of the pathway, other important parameters must be considered, such as: 1- intracellular TCS protein concentrations and relative abundance, both of which might change with time; 2- potential subcellular localization (spatial segregation of HK and/or RR), which could also change with time in a signal-dependent manner; 3- RR dimerization/oligomerization, which could affect HK-binding in different ways according to variant quaternary structure organizations; 4- presence of dedicated P~RR phosphatases; 5- RR autophosphorylation with intracellular small phosphodonors; and probably many others. All of these parameters can easily be linked to the detailed time-course of appearance and disappearance of HK and RR phosphorylated species in the cell, independent of any structural determinants within the HK and RR molecules themselves.

*4) The authors missed some opportunities for more effective communication of key results. In particular, the mechanism of phosphatase activity by HK remains unresolved and whether or not the conserved His residue participates in the reaction is controversial. For example, a recent prominent review article [Bhate, Structure 23, 981 (2015)] states that although the His is not always required, "in cases where the catalytic histidine is required, it likely acts as a base to assist the attack of water or hydroxide on the phosphoryl-aspartyl group". The findings reported in this manuscript place the His far from the attacking water.*

We take the point and thank the reviewers for the suggestion. We have further elaborated our findings concerning the phosphatase complex, in the Results subsection entitled “The phosphatase to phosphotransferase transition: reconfiguring the reaction center for catalysis”. In particular, we embed our interpretations within the context of reports concerning phosphatase mechanisms postulated for HisKA and HisKA_3 HKs.

Bhate et al., 2015 indeed refer to HKs that use the reactive His during the P~RR dephosphorylation reaction. We have now added this reference, we agree it represents one of the most recent reviews in the field and a very helpful one. However, for the specific matter being discussed, this review doesn’t really refer to any original papers showing experimental evidence in favor of such catalytic role of the His in the phosphatase reaction. There is of course the early paper coming from Inouye’s team, claiming such a role (Zhu et al., 2000; now also referenced), although it should be stressed that it is contradictory to previous reports studying the same his kinase, EnvZ (e.g. Skarphol et al., 1997; or yet Hsing & Silhavy 1997): perhaps the fact of using separate DHp and CA domains in the PNAS article might have led to equivocal readouts.

In contrast, several articles present converging evidence in support of a different residue playing a key role in mediating dephosphorylation: typically an amide side chain, one helical turn away of the phosphorylatable His, which has been proposed to position/assist a catalytic water to perform the nucleophilic attack onto the leaving phosphoryl group. We have thus also referred now to a few reports (Huynh et al., 2010; Willett & Kirby, 2012; Skarphol et al., 1997) that show nonessentiality of the His, and instead a key role for the amide one helical turn C-terminal to it. As also shown in the phosphatase CheX complexed to CheY3 (Pazy et al., 2010), and further consistent with the CheZ:CheY (Zhao et al., 2002) and the RapH-Spo0F (Parashar et al., 2011) complex structures.

As we now describe more extensively, the phosphatase structure we now report is clearly favoring the latter hypothesis. In DesK it is Gln193, 5 residues C-terminal to the reactive His, poised to performing the water-assisting role. DesKQ193 is homologous to Gln404 in NarX, extensively studied by V Stewart and collaborators in the 2010 PNAS paper mentioned above, shown to be essential for phosphatase activity. Particularly illustrative is Figure 1 in that article, to pinpoint equivalent amide-containing amino acids at that position in different HK families and dedicated phosphatases: the HDxxxQ motif of HisKA_3 HKs such as NarX and DesK (where H is the phosphorylatable His, and Q is the phosphatase-essential Gln), corresponds to a H[E/D]xx[T/N] in HisKA HKs (note one residue less in between the His and the amide; and the only case where a number of members display a Thr, instead of an amide side chain), or yet to motifs DxxxQ and ExxN in phosphatases like CheZ and CheX, with the amide-containing residue occupying equivalent structural positions when superposed onto the HKs.

*5) Similarly, for readers interested in mechanistic details, it would be helpful to include illustrations of the proposed active sites for the phosphotransfer and phosphatase reactions showing just the relevant side chains and small molecules, and indicating distances and geometries, without the clutter of backbone ribbons.*

We have now added these two panels as an additional figure supplement to Figure 4.

*6) There is great merit in providing a proposed mechanism such as in Figure 6, which provides specific features that can be tested. I have two comments about the illustration and accompanying text:a) In subsection “A HK:RR concerted switch controls efficiency along the signaling pathway” the authors say that a hydrogen bond is formed between His188 and Asp189. Is this modeled in the reported structures or just speculation? The reference cited (Quezada et al. 2005) actually demonstrated hydrogen bonding between the conserved His and a Glu that is 22 residues to the C-terminal side, not 1 residue C-terminal as in DesK. It might be worth citing Willett & Kirby, PLOS Genetics 8, e1-003984 (2012), which provides evidence in multiple HKs that the acidic residue at variable position H^+^1 is critical for autophosphorylation, but not for phosphatase activity.*

This hydrogen bond is expected to be present during autophosphorylation, hence not present in our reported structures. Both Casino et al. 2014, and Mechaly et al. 2014 papers, reporting autophosphorylating HKs, do observe this H-bond. The Quezada et al. 2005 article is referenced there, because even when the carboxylate group in that case (CheA) is indeed coming from a completely different place (compared to the more canonical DHp-containing HKs), it is a very nice paper where direct NMR spectroscopy data reveal how that carboxylate H-bond stabilizes the reactive protonation tautomer of the reactive His during autophosphorylation.

We regret the initial phrasing was not clear enough. We have now rephrased that fragment, hoping it is now clearer. We also agree that adding the Willett & Kirby, 2012 reference is useful to further support the critical function of a carboxylate-containing side chain at position His+1 for autophosphorylation.

*b) The proposed dissociative phosphotransfer mechanism involves a key interaction between the conserved Ser/Thr residue of the RR and the phosphoryl group that has been released from the HK, before attack by the RR Asp. One prediction of this mechanism might be that phosphotransfer from HK to RR would be severely affected in the absence of the conserved Ser/Thr residue. I am aware of only one potentially relevant piece of data in the literature. Figure 3 of Appleby & Bourret, J Bacteriol 180, 3563 (1998) shows that phosphotransfer occurs from CheA to a CheY mutant in which the conserved Thr is repaced by an Ala. Obvious caveats are that CheAs are different than other HKs, and that the experiment was not particularly sensitive to changes in phosphotransfer rate (i.e. the rate of phosphotransfer could have been dramatically slower but still observed).*

Indeed, our model predicts that Thr80 might play a key role in HK-mediated phosphotransfer, and probably increasingly so according to the degree of dissociative character of the phosphoryl-transfer reaction. We have now included a few words to make this point clearer for the readers. In the Discussion subsection “HK:RR concerted switch controls efficiency along the signaling pathway.”, when explaining the conceptual model (Figure 6), we have now made this explicit, awaiting for direct and conclusive proof.

In favor of a key role of this Thr in the reactions that RRs undertake, it is an extremely well conserved residue at that position (at times replaced by a Ser, prone to play a similar role). But, we are aware of the paper you are citing (Appleby and Bourret, 1998), even taking into account the caveats associated to CheA’s singularities, is strongly suggesting that Thr80 (T87 in CheY) is not playing a catalytic role.

Surprisingly enough, very few papers actually interrogate the phosphoryl transfer activity substituting this ultra-conserved amino acid, in order to have a more solid background about its role. There is Ganguli et al., 1995, which had previously studied also CheY. And beyond CheA:CheY, Gautam et al. 2011 report data on DevR (*aka* DosR) from *M. Tuberculosis*, where they substitute the equivalent Thr82. Although both Ganguli and Gautam articles detect HK-mediated phosphotransfer activity in vitro, using a Thr-to-Ala substituted RR, they do report a significant decrease in such transfers. All these papers, including Appleby & Bourret, observe substantial perturbations and even disruption of in vivosignaling when this Thr is substituted (and, consistently, Thr-to-Ser giving milder or undetectable deleterious effects). Taking those pieces of evidence together, it seems that this Thr is not a catalytic residue *sensu strictu* (i.e. one that directly participates in the chemistry within the reaction center, such that its substitution results in a drop of reaction velocity of several orders of magnitude, in practical terms usually to undetectable limits for in vitroenzymatic assays), but one that plays a major role in the reaction, detectable in vitro, and to the point of precluding normal biological function.

There are a number of positions that, derived from the analysis of the DesKC:DesR-REC structures, seem extremely interesting to further analyze in-depth, among others Thr80 in the RR, and also key positions in the HK component. The functional and structural consequences of such critical residue substitutions will be published in a separate paper. At this point, taking now into account the crystallographic evidence that we are reporting, it is tempting to speculate that Thr80 might play an increasingly essential role in phosphoryl-transfer, correlated to the increasing degree of dissociative character of such reaction for a given TCS pair.

In sum, the receiver Asp residue plays the catalytic role in all cases, performing the nucleophilic attack on the migrating phosphorus atom. If the His(HK)-to-Asp(RR) distance is short, the transfer is largely associative, and Thr80 might play less essential roles e.g. in further stabilizing ground states (e.g. modulating P~Asp stability). But, for TCS complexes where that His(HK)-to-Asp(RR) distance is large, such as in DesK:DesR and others, nucleophilic substitution chemistry predicts a non-negligible amount of free metaphosphate intermediate for the reaction to occur, and it is in this case where the conserved Thr is expected to play a more essential role (again, maybe not catalytic *sensu strictu*, but without which the velocity could be reduced substantially because of insufficient intermediate state stabilization – the limit of catalytic vs non catalytic becoming of course increasingly ambiguous).

*7) The apparently important role played in the reactions by variable RR residues is very interesting. For example, subsection “The phosphatase to phosphotransferase transition: reconfiguring the reaction center for catalysis” notes apparent hydrogen bonding between the phosphoryl group and Thr81 of DesK. One wonders what happens in the ~80% of RRs for which the residue corresponding to Thr81 is not a Ser or Thr. Similarly, in this section and elsewhere, the role of Phe82 in forming the phosphate lid is noted. One wonders what happens in the 2/3 of response regulators that do not contain an aromatic residue (Tyr, Phe, His) at the position corresponding to Phe82. These are rich topics for future investigation. It might be worth noting someplace in the manuscript that the reported structures suggest important roles for variable residues, which by definition vary between systems.*

This is certainly an interesting point, we have added a paragraph towards the end of the Discussion subsection “HK:RR concerted switch controls efficiency along the signaling pathway”, highlighting the importance of some of these variable residues. Several of them had already been pinpointed by studying standalone RR catalytic activities (auto-phosphorylation and auto-dephosphoryation), mainly in CheY, but more recently also in RRs belonging to other families. We have thus added a few key references.

As mentioned now in the revised version of the text, we believe that the evolution of the HK:RR interface adds a new dimension in terms of selective pressure constraints, acting on several of these highly variable positions. Future studies focusing on HK-mediated phosphorylation/dephosphorylation of RRs with substitutions on these positions, might reveal covariation constraints (epistasis again, but now between both protein partners), perhaps further explaining population distributions among RR families like the one summarized in Table 3 of Page et al., 2016.